# Convergence Properties of Natural Gradient Descent for Minimizing KL Divergence

**Adwait Datar**[1]                                                                    *adwait.datar@tuhh.de*

**Nihat Ay**[1,2,3]                                                                    *nihat.ay@tuhh.de*

[1] *Institute for Data Science Foundations*
*Hamburg University of Technology*
*21073 Hamburg, Germany*

[2] *Santa Fe Institute*
*Santa Fe, NM 87501, USA*

[3] *Leipzig University*
*04109 Leipzig, Germany*

**Reviewed on OpenReview:** *https://openreview.net/forum?id=h6hjjAF5Bj*

## Abstract

The Kullback-Leibler (KL) divergence plays a central role in probabilistic machine learning, where it commonly serves as the canonical loss function. Optimization in such settings is often performed over the probability simplex, where the choice of parameterization significantly impacts convergence. In this work, we study the problem of minimizing the KL divergence and analyze the behavior of gradient-based optimization algorithms under two dual coordinate systems within the framework of information geometry— the exponential family ($\theta$ coordinates) and the mixture family ($\eta$ coordinates). We compare Euclidean gradient descent (GD) in these coordinates with the coordinate-invariant natural gradient descent (NGD), where the natural gradient is a Riemannian gradient that incorporates the intrinsic geometry of the underlying statistical model. In continuous time, we prove that the convergence rates of GD in the $\theta$ and $\eta$ coordinates provide lower and upper bounds, respectively, on the convergence rate of NGD. Moreover, under affine reparameterizations of the dual coordinates, the convergence rates of GD in $\eta$ and $\theta$ coordinates can be scaled to $2c$ and $\frac{2}{c}$, respectively, for any $c > 0$, while NGD maintains a fixed convergence rate of 2, remaining invariant to such transformations and sandwiched between them. Although this suggests that NGD may not exhibit uniformly superior convergence in continuous time, we demonstrate that its advantages become pronounced in discrete time, where it achieves faster convergence and greater robustness to noise, outperforming GD. Our analysis hinges on bounding the spectrum and condition number of the Hessian of the KL divergence at the optimum, which coincides with the Fisher information matrix.

## 1 Introduction

The convergence properties of the natural gradient descent algorithm, originally introduced in Amari (1996), have been extensively studied in the literature (e.g., Amari (1998); Pascanu & Bengio (2014); Martens (2020)). In particular, the natural policy gradient Kakade (2001) has motivated a rich body of research (see, e.g., Müller & Montúfar (2024); Yuan et al. (2022); Khodadadian et al. (2022).) Beyond this, the natural gradient methods have been applied to diverse problems including Bayesian networks Ay (2023); Ay & van

Oostrum (2023), over-parametrized neural networks Zhang et al. (2019); van Oostrum & Ay (2021); van Oostrum et al. (2023) and infinitely-wide networks Karakida et al. (2019); Karakida & Osawa (2021) to name a few. Related to our focus, recent work has also investigated convergence rates of natural gradient flows and their discrete counterparts (see Zhang et al. (2019); Xiao (2022); Yuan et al. (2022); Khodadadian et al. (2022); Müller & Montúfar (2024)). A commonly observed phenomenon is that natural gradient descent outperforms Euclidean gradient descent, albeit at a higher computational cost. In this work, we revisit this comparison in a simple yet illuminating setting: minimizing the Kullback-Leibler (KL) divergence over the probability simplex. The KL divergence is a fundamental loss function in probabilistic machine learning, arising naturally from the maximum likelihood principle and information-theoretic considerations (Mohri et al., 2018, Section 12.1.1). Despite the apparent simplicity of the problem, we observe that natural gradient flows do not universally outperform standard Euclidean gradient flows.

Specifically, we consider two dual parametrizations of the probability simplex: the exponential family representation (the $\theta$ coordinates) and the mixture family representation (the $\eta$ coordinates) Amari (2016). We prove that the natural gradient flow converges faster than the Euclidean gradient flow in the $\theta$ coordinates (the $\theta$-*gradient flow*), consistent with results in the literature. However, the natural gradient flow (despite yielding straight-line trajectories) converges more slowly than the Euclidean gradient flow in $\eta$ coordinates ($\eta-$gradient flow). This demonstrates that the often-reported rapid convergence of natural gradient flow cannot be simplistically attributed to the straightness of its trajectories. Furthermore, leveraging the invariance of the canonical divergence under affine transformations, we show that the convergence rates of the Euclidean gradient flows in $\eta$ and $\theta$ coordinates can be adjusted to $2c$ and $\frac{2}{c}$, respectively, for an arbitrary $c > 0$, while the natural gradient maintains a fixed convergence rate of 2, sandwiched between them. Thus, by setting $c = 1$, we can construct a pair of dual coordinates, $(\bar{\eta}, \bar{\theta})$, that match the convergence rate of the natural gradient flow. Since the advantages of the natural gradient are not immediately apparent in the continuous-time setting, we extend our analysis to the discrete-time case, where the natural gradient demonstrates both faster convergence and greater robustness to noise, outperforming Euclidean gradient descent. We show that the fundamental reason behind the superiority of natural gradient lies in the optimal conditioning of the loss landscape: the natural gradient updates are equivalent to minimizing the loss function $\frac{1}{2}\|\eta - \eta_q\|^2$, whose Hessian has a condition number equal to 1. The main contributions of this paper are summarized as follows:

1. We analyze the convergence rates of Euclidean gradient flows in $\eta$ and $\theta$ coordinates, and of the natural gradient flow (Theorem 3 and Theorem 9). We show that while the natural gradient flow converges faster than the $\theta$-gradient flow, it is slower than the $\eta$-gradient flow. This result builds upon bounds on the spectrum of the Hessian of the loss function established in Lemma 2. These theoretical findings are supported by illustrative numerical experiments.

2. Exploiting the duality and the invariance of the canonical divergence under affine transformations, we demonstrate in Theorem 4 that the convergence rates of Euclidean gradient flows in $\eta$ and $\theta$ coordinates can be adjusted to $2c$ and $\frac{2}{c}$, respectively, for an arbitrary $c > 0$, while the natural gradient maintains a fixed convergence rate of 2, sandwiched between them. This shows that there exists a pair of dual coordinates, $(\bar{\eta}, \bar{\theta})$ such that the convergence rate of $\bar{\eta}-$ and $\bar{\theta}-$gradient flows matches the convergence rate of the natural gradient flow.

3. We analyze the discrete-time dynamics in Section 4, where Theorems 7 and 8 establish the superior robustness properties of natural gradient dynamics compared to their Euclidean counterparts. The core reason for this superiority is the optimal conditioning of the underlying loss landscape. In particular, natural gradient updates can be interpreted as minimizing the loss function $\frac{1}{2}\|\eta - \eta_q\|^2$, whose Hessian exhibits the ideal condition number of 1.

4. We complement our theoretical results with empirical studies in Section 5, extending the analysis to practical settings where only finite samples from the target distribution are available. Specifically, we consider optimization of the empirical KL divergence in both full-batch and stochastic gradient descent (SGD) settings. Our results show that NGD consistently outperforms standard GD when learning rates are optimally tuned, in alignment with Theorems 5, 7, and 8. Furthermore, when using sufficiently small and equal learning rates across all methods, the sandwiching behavior predicted

by Theorem 3 for continuous-time dynamics persists in the discrete-time, sample-based setting thus validating the relevance of our theory beyond idealized assumptions.

**Notation**

Let $\mathbb{R}$ denote the set of real numbers. For a function $g$ of two variables $x \in \mathbb{R}^n$ and $y \in \mathbb{R}^m$, $g : \mathbb{R}^n \times \mathbb{R}^m \to \mathbb{R}$, we use the notation

$$
\nabla_x g(x,y) = \begin{bmatrix} \frac{\partial g}{\partial x_1}(x,y) \\ \vdots \\ \frac{\partial g}{\partial x_n}(x,y) \end{bmatrix}, \quad \nabla_x^2 g(x,y) = \begin{bmatrix} \frac{\partial^2 g}{\partial x_1^2}(x,y) & \cdots & \frac{\partial^2 g}{\partial x_1 \partial x_n}(x,y) \\ \vdots & \ddots & \vdots \\ \frac{\partial^2 g}{\partial x_n \partial x_1}(x,y) & \cdots & \frac{\partial^2 g}{\partial x_n^2}(x,y) \end{bmatrix}.
$$

For functions $f$ of a single variable $x$, we suppress the subscript and simply write $\nabla f(x)$ and $\nabla^2 f(x)$. For a manifold $\mathcal{M}$ with two global charts $\phi_m : \mathcal{M} \to \phi_m(\mathcal{M}) \subset \mathbb{R}^n$ and $\phi_e : \mathcal{M} \to \phi_e(\mathcal{M}) \subset \mathbb{R}^n$ with coordinates $\eta \in \phi_m(\mathcal{M})$ and $\theta \in \phi_e(\mathcal{M})$, we slightly abuse notation and write for any smooth function $\mathcal{L} : \mathcal{M} \to \mathbb{R}$,

$$
\begin{aligned}
\mathcal{L}(\eta) &:= \mathcal{L}(\phi_m^{-1}(\eta)), & \mathcal{L}(\theta) &:= \mathcal{L}(\phi_e^{-1}(\theta)), \\
\nabla \mathcal{L}(\eta) &:= \nabla \left( \mathcal{L} \circ \phi_m^{-1} \right)(\eta), & \nabla \mathcal{L}(\theta) &:= \mathcal{L}(\phi_e^{-1}(\theta)), \\
\nabla^2 \mathcal{L}(\eta) &:= \nabla^2 \left( \mathcal{L} \circ \phi_m^{-1} \right)(\eta), & \nabla^2 \mathcal{L}(\theta) &:= \nabla^2 \left( \mathcal{L} \circ \phi_e^{-1} \right)(\theta).
\end{aligned}
$$

For a point $p \in \mathcal{M}$, we write $\eta_p = \phi_m(p)$ and $\theta_p = \phi_e(p)$ to denote the point $p$ in the $\eta$ and $\theta$ coordinates, respectively. For a symmetric matrix $Q$, we write $Q \succ 0$ (resp. $Q \succeq 0$) to denote that $Q$ is symmetric positive definite (resp. positive semi-definite). Building on this notation, we write $Q \succ P$ (resp. $Q \succeq P$) to mean $Q - P \succ 0$ (resp. $Q - P \succeq 0$). For a symmetric matrix $Q$, let $\lambda_{\min}(Q)$ and $\lambda_{\max}(Q)$ denote the minimum and maximum eigenvalue of $Q$. Since $Q \succ 0$ implies that all eigenvalues of $Q$ are positive, we can define the condition number of $Q$ as $\text{cond}(Q) := \frac{\lambda_{\max}(Q)}{\lambda_{\min}(Q)}$. For any $x \in \mathbb{R}^n$, let $\|x\|$ denote the standard Euclidean norm, and define the closed norm ball of radius $\varepsilon$ centered at $x$ by $\mathcal{B}_\varepsilon(x) := \{y \in \mathbb{R}^n : \|y - x\| \leq \varepsilon\}$. For any real matrix $M$, let $\|M\|_2 := \sup_{x \neq 0} \frac{\|Mx\|}{\|x\|}$ be the induced matrix 2-norm, which coincides with the maximum eigenvalue of $M$ when $M \succ 0$.

## 2 Information Geometry Preliminaries and Gradient Flow Dynamics

In this section, we review the information geometric preliminaries and arrive at the continuous-time gradient flow dynamics which are then analyzed in the following section. For further details on the underlying concepts of information geometry, the reader is referred to Amari (2016); Amari & Nagaoka (2000); Ay et al. (2017).

### 2.1 Discrete Distributions in Mixture and Exponential Coordinates

Consider the family $S_n$ of probability distributions over a discrete random variable $X$ with sample space $\Omega = \{1, 2, \cdots, n, n+1\}$. Let $p_i$ be the probability that $X$ takes the value $i$. Then, any $p \in S_n$ can be written as

$$
p(x) = \sum_{i=1}^{n+1} p_i \delta_i(x),
$$

where $\delta_i(x)$ is the delta distribution over $\Omega$, concentrated at $i$. Thus, $S_n$ can be identified with the $n-$dimensional simplex[1], i.e., $S_n = \left\{ (p_1, p_2, \cdots, p_n, p_{n+1}) \in \mathbb{R}^{n+1} \mid p_i > 0, \sum_{i=1}^{n+1} p_i = 1 \right\}$. This family admits representations both as a mixture family and an exponential family Amari (2016). This can be seen by noticing that any $p \in S_n$ can be written as

$$
p(x) = \underbrace{\sum_{i=1}^{n} \eta_i \delta_i(x) + \left( 1 - \sum_{k=1}^{n} \eta_k \right) \delta_{n+1}(x)}_{\text{Mixture family representation}} = \underbrace{\exp \left( -\psi(\theta) + \sum_{i=1}^{n} \theta_i \delta_i(x) \right)}_{\text{Exponential family representation}},
$$

---

[1]Note that our definition of the simplex excludes the boundary.

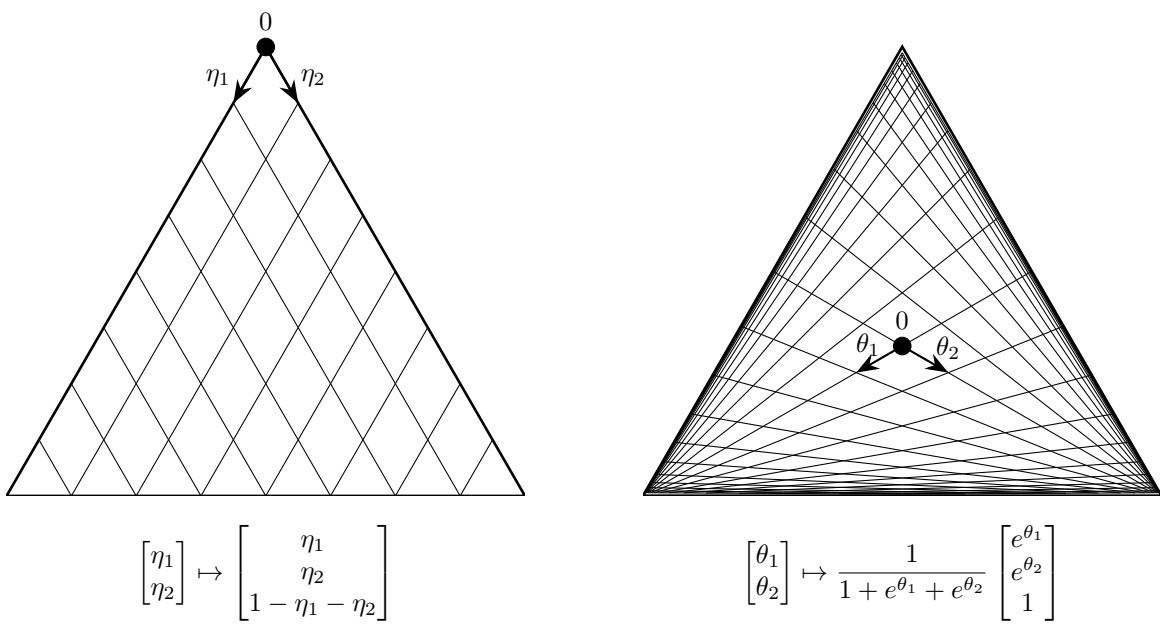

Figure 1: Left: Coordinate system with the natural parameters $\eta = (\eta_1, \eta_2)$ of the mixture family representation of $S_2$. Right: Coordinate system with the natural parameters $\theta = (\theta_1, \theta_2)$ of the exponential family representation of $S_2$.

where $\psi(\theta) = \log\left(1 + \sum_{i=1}^{n} e^{\theta_i}\right)$ is the log-partition function ensuring the normalization constraint $\sum_{x \in \Omega} p_\theta(x) = 1$ for the exponential family representation. With $\eta = (\eta_1, \eta_2, \cdots, \eta_n)$, we obtain a coordinate system for the simplex, with $\eta$ serving as the natural parameter of the mixture family. We let $\phi_m : S_n \ni p \mapsto \eta = (\eta_1, \eta_2, \cdots, \eta_n) \in \mathbb{R}^n$ denote the global chart for $S_n$ in the mixture family coordinate system. This is depicted in Fig. 1 (left). Similarly, with $\theta := (\theta_1, \theta_2, \cdots, \theta_n)$, we obtain an alternate coordinate system for the simplex with $\theta$ being the natural parameter of the exponential family. Define $\phi_e : S_n \ni p \mapsto \theta = (\theta_1, \theta_2, \cdots, \theta_n) \in \mathbb{R}^n$ as the global chart for $S_n$ in the exponential family coordinate system. This is depicted in Fig. 1 (right).

## 2.2 Convex Duality and Bregman Divergence

For the family $S_n$, there exists a dual relationship between the coordinates $\eta$ and $\theta$. Since $\psi(\theta) = \log\left(1 + \sum_{i=1}^{n} e^{\theta_i}\right)$ is strictly convex, it is possible to define its convex conjugate $\varphi(\eta) = \max_\vartheta \left(\eta^T \vartheta - \psi(\vartheta)\right)$. Optimality condition on the maximizer $\vartheta_{\text{opt}} = \theta$ yields the relationship $\nabla \psi(\theta) = \eta$, which can be solved to obtain $\theta_i = \log\left(\frac{\eta_i}{1 - \sum_{i=1}^{n} \eta_i}\right)$. This results in $\varphi(\eta) = \left(\eta^T \theta - \psi(\theta)\right) = \sum_{i=1}^{n+1} \eta_i \log \eta_i$, the negative of Shannon entropy. Conversely, $\psi$ is the convex conjugate of $\varphi$ leading to $\theta = \nabla \varphi(\eta)$. Since $\nabla_\eta \varphi(\nabla_\theta \psi(\cdot))$ is the identity map, application of the chain rule gives

$$\nabla^2 \varphi(\eta) = \left[\nabla^2 \psi(\theta)\right]^{-1}. \tag{1}$$

The convex conjugate functions $\psi$ and $\varphi$ define a pair of Bregman Divergence $D_\psi$ and $D_\varphi$ satisfying

$$D_\psi(\theta_p \parallel \theta_q) := \psi(\theta_p) - \psi(\theta_q) - \nabla\psi(\theta_q)^T (\theta_p - \theta_q) \tag{2}$$

$$= \varphi(\eta_q) - \varphi(\eta_p) - \nabla\varphi(\eta_p)^T (\eta_q - \eta_p) =: D_\varphi(\eta_q \parallel \eta_p). \tag{3}$$

In our setting of $S_n$, this Bregman divergence equals the canonical KL-divergence $D(q||p)$ between probability distributions $q$ and $p$, i.e.,

$$D_\psi(\theta_p \parallel \theta_q) = D_\varphi(\eta_q \parallel \eta_p) = D(q||p) = \sum_{i=1}^{n+1} q_i \log\left(\frac{q_i}{p_i}\right), \tag{4}$$

where $(\eta_q, \eta_p)$ and $(\theta_q, \theta_p)$ are the coordinate representations of $(q, p)$ in the $\eta$ and $\theta$ coordinates, respectively. For further details, the reader is referred to Amari & Nagaoka (2000).

## 2.3 Gradient and Natural Gradient Dynamics

For a given target distribution $q \in S_n$, let the loss function $\mathcal{L}_q : S_n \to \mathbb{R}$ be defined by $\mathcal{L}_q(p) = D(q||p)$. As discussed in Section 1, we abuse the notation slightly and interchangeably use $q$, $\theta_q = \phi_e(q)$ or $\eta_q = \phi_m(q)$ to denote probability distribution $q \in S_n$, the same distribution in $\theta$ coordinates and in $\eta$ coordinates, respectively. Thus, we write $\mathcal{L}_q(\eta_p)$ to mean $\mathcal{L}_q(\phi_m^{-1}(\eta_p))$ and $\mathcal{L}_q(\theta_p)$ to mean $\mathcal{L}_q(\phi_e^{-1}(\theta_p))$. The gradient of the loss function can be computed in the different coordinate systems using equation 2 and equation 4 as

$$\nabla\mathcal{L}_q(\eta_p) = -\nabla\varphi(\eta_p) - \nabla^2\varphi(\eta_p)(\eta_q - \eta_p) + \nabla\varphi(\eta_p) = -\nabla^2\varphi(\eta_p)(\eta_q - \eta_p), \tag{5}$$

$$\nabla\mathcal{L}_q(\theta_p) = \nabla\psi(\theta_p) - \nabla\psi(\theta_q). \tag{6}$$

Analogously, for a given target distribution $p \in S_n$, let the loss function $\mathcal{L}_p^* : S_n \to \mathbb{R}$ be defined by $\mathcal{L}_p^*(q) = D(q||p)$. The gradient of this loss function can be computed in the different coordinate systems using equation 2 and equation 4 as

$$\nabla\mathcal{L}_p^*(\eta_q) = \nabla\varphi(\eta_q) - \nabla\varphi(\eta_p), \tag{7}$$

$$\nabla\mathcal{L}_p^*(\theta_q) = -\nabla\psi(\theta_q) - \nabla^2\psi(\theta_q)(\theta_p - \theta_q) + \nabla\psi(\theta_q) = -\nabla^2\psi(\theta_q)(\theta_p - \theta_q). \tag{8}$$

Building on the pair of conjugate dual functions, we define a Riemannian metric $g$ on $S_n$, which assigns to each point $p \in S_n$ an inner product $\langle\cdot,\cdot\rangle_p$ on the tangent space $T_pS_n$. The tangent space $T_pS_n$ can be identified with $\left\{(v_1, v_2, \cdots, v_n, v_{n+1}) \in \mathbb{R}^{n+1} | \sum_{i=1}^{n+1} v_i = 0\right\}$ (see Ay et al. (2017)). In coordinates, this Riemannian metric is represented by the matrix $\nabla^2\varphi(\eta)$ in the $\eta$ coordinates and $\nabla^2\psi(\theta)$ in the $\theta$ coordinates. Importantly, this Riemannian metric coincides with the Fisher metric (Amari, 2016, Theorem 2.1). Using this Riemannian metric, we define the Riemannian gradient grad $\mathcal{L}_q(p)$, also called as the natural gradient, at a point $p \in S_n$ through the relation

$$\langle\text{grad } \mathcal{L}_q(p), v\rangle_p = d\mathcal{L}_q(p)[v] \tag{9}$$

for all $v$ in the tangent space of $S_n$ at the base point $p$. This defining property allows us to compute the natural gradient in $\eta$ coordinates using equation 5 as

$$\text{grad } \mathcal{L}_q(\eta_p) = \left[\nabla^2\varphi(\eta_p)\right]^{-1}\nabla\mathcal{L}_q(\eta_p) = -(\eta_q - \eta_p).$$

Similarly, grad $\mathcal{L}_p^*(q)$ can be computed in the $\theta$ coordinates using equation 8 as grad $\mathcal{L}_p^*(\theta_q) = -(\theta_p - \theta_q)$. Interestingly, the natural gradients when represented in appropriate coordinates take on particularly simple linear forms − they directly point towards the target distributions. Since the situation with the loss function $\mathcal{L}_p^*$ is analogous to that of $\mathcal{L}_q$, for brevity, we will focus our analysis on $\mathcal{L}_q$ for the remainder of the paper[2].

We now introduce the gradient flow dynamics in both $\eta$ and $\theta$ coordinates, as well as the natural gradient flow which will be analyzed in the subsequent sections. Although the natural gradient flow dynamics are coordinate-invariant, we express them in $\eta$ coordinates to exploit the particularly simple linear structure.

For a given target distribution $q \in S_n$ and an initial distribution $p_0 \in S_n$, consider the gradient flow dynamics described by equation 10 and equation 11 and the natural gradient flow dynamics described by equation 12

$$\dot{\eta}(t) = -\nabla\mathcal{L}_q(\eta(t)), \qquad \eta(0) = \eta_{p_0}, \tag{10}$$

$$\dot{\theta}(t) = -\nabla\mathcal{L}_q(\theta(t)), \qquad \theta(0) = \theta_{p_0}, \tag{11}$$

$$\dot{\eta}_{ng}(t) = -\text{grad } \mathcal{L}_q(\eta_{ng}(t)), \quad \eta_{ng}(0) = \eta_{p_0}. \tag{12}$$

---

[2]We include the convergence rate analysis of gradient flows for $\mathcal{L}_p^*$ in Appendix A for completeness.

We will analyze these dynamics in the following sections.

## 3  Convergence Analysis in Continous Time

We start the convergence analysis with a general result which is at the core of the analysis. This result frequently appears in various forms, typically emphasizing the upper bound in inequality 14 (see (Wensing & Slotine, 2020, Proposition 1) for example). We present an adaptation of this result to our setting.

**Proposition 1** (Convergence of general gradient flows). *Consider the gradient flow dynamics*

$$\dot{x}(t) = -\nabla f(x(t)), \quad x(t_0) = x_0, \tag{13}$$

*where a sufficiently smooth function $f : U \to \mathbb{R}$, with $U \subset \mathbb{R}^n$ being an open neighborhood of $x_0$, satisfies the following properties:*

(a) *$f$ has a unique minimizer $x_* \in U$ satisfying $\nabla f(x_*) = 0$.*

(b) *there exist positive constants $m$ and $L$ such that $m \cdot I \preceq \nabla^2 f(x) \preceq L \cdot I$ for all $x$ in the sublevel set $S := \{x \in U | f(x) \leq f(x_0)\}$.*

*Then the solution $x : [t_0, \infty) \to U$ of equation 13 satisfies*

(i) *$x(t) \in S$ for all $t \geq t_0$ and*

(ii) *With $c = f(x_0) - f(x_*)$, we get that*

$$c \cdot e^{-2L(t-t_0)} \leq f(x(t)) - f(x_*) \leq c \cdot e^{-2m(t-t_0)} \text{ for all } t \geq t_0, \tag{14}$$

*i.e., $f(x(t))$ converges exponentially to $f(x_*)$ with a rate larger than $2m$ and smaller than $2L$.*

*Proof.* See Appendix C.1 □

To facilitate the application of Proposition 1 to the dynamics given in equation 10 and equation 11, we establish bounds on the Hessian of the loss function in the following Lemma.

**Lemma 2** (Bounds on the Hessian of the loss function). *Let $p, q \in S_n$. Then the following statements hold:*

(i) *(Global bound) The Hessians of $\mathcal{L}_q$ and $\mathcal{L}_p^*$ satisfy*

$$0 \prec \nabla^2 \mathcal{L}_q(\theta) \prec I \prec \nabla^2 \mathcal{L}_p^*(\eta) \qquad \forall \, \theta \in \phi_e(S_n), \forall \, \eta \in \phi_m(S_n) \qquad \text{and} \tag{15}$$
$$0 \prec \nabla^2 \mathcal{L}_q(\eta) \qquad \qquad \forall \, \eta \in \phi_m(S_n). \tag{16}$$

(ii) *(Local bound at optimum) The Hessians of $\mathcal{L}_q$ and $\mathcal{L}_p^*$ evaluated at the optimum satisfy*

$$I \prec \nabla^2 \mathcal{L}_q(\eta_q) = \nabla^2 \mathcal{L}_q(\theta_q)^{-1}, \tag{17}$$
$$I \prec \nabla^2 \mathcal{L}_p^*(\eta_p) = \nabla^2 \mathcal{L}_p^*(\theta_p)^{-1}, \tag{18}$$

*Proof.* See Appendix C.2 □

The positive definiteness of the Hessians established in Lemma 2 (see equation 15 and equation 16) shows that $\mathcal{L}_q$ is convex in the $\theta$ coordinates as well as in the $\eta$ coordinates, and $\mathcal{L}_p^*$ is convex in the $\eta$ coordinates. One might conjecture that $\mathcal{L}_p^*$ might likewise be convex in the $\theta$ coordinates. However, this turns out not to be the case, as illustrated by a counterexample. Figure 2 shows a sample contour plot of $\mathcal{L}_p^*(\theta)$ for $n = 2$. The plot clearly shows that the sublevel sets of $\mathcal{L}_p^*(\theta)$ are non-convex, disproving the conjecture. To summarize, while the KL divergence is geodesically convex along $m$-geodesics in both of its arguments, it is geodesically

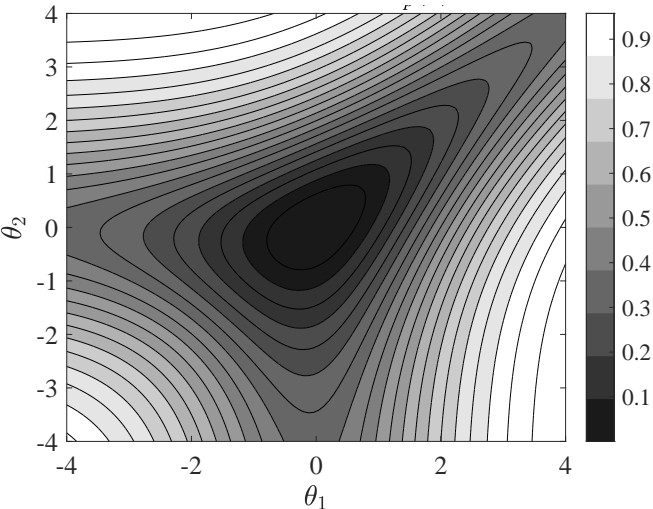

Figure 2: Contour plot of $\mathcal{L}_p^*(\theta)$ showing that the sublevel sets are non-convex.

convex along $e$-geodesics only with respect to its second argument (see (Boumal, 2023, Definition 11.3) for definition of geodesic convexity and (Amari, 2016, Section 2.4) for definitions of $e-$ and $m-$ geodesics). With these uniform bounds on the Hessians in place, we are now equipped to control the exponential decay rates of gradient flows through Proposition 1. This is presented in the next result which is the main result of this section.

**Theorem 3** (Convergence analysis). *Let $q \in S_n$ be the target distribution and $p_0 \in S_n$ be the initial distribution. Suppose $\eta$, $\theta$ and $\eta_{ng}$ be the solutions to dynamics described by equation 10, equation 11 and equation 12, respectively. Then*

(i) *there exist positive constants $1 < m_\eta \leq L_\eta$, $c_\eta$, $\bar{c}_\eta$ and $T$ such that*

$$c_\eta e^{-2L_\eta t} \leq \mathcal{L}_q(\eta(t)) \leq \bar{c}_\eta e^{-2m_\eta t} \leq \bar{c}_\eta e^{-2t} \qquad \forall t \geq T \tag{19}$$

*i.e., $\mathcal{L}_q(\eta(t))$ converges to zero exponentially with rate higher than 2. Furthermore, if $\mathcal{L}_q(\eta(0))$ is sufficiently small, the result holds with $T = 0$.*

(ii) *there exist positive constants $m_\theta \leq L_\theta < 1$ and $c_\theta$ such that*

$$c_\theta e^{-2t} \leq c_\theta e^{-2L_\theta t} \leq \mathcal{L}_q(\theta(t)) \leq c_\theta e^{-2m_\theta t} \qquad \forall t \geq 0, \tag{20}$$

*i.e., $\mathcal{L}_q(\theta(t))$ converges to zero exponentially with rate lower than 2.*

(iii) *there exist positive constants $c_1$ and $c_2$ such that*

$$c_1 e^{-2t} \leq \mathcal{L}_q(\eta_{ng}(t)) \leq c_2 e^{-2t} \qquad \forall t \geq 0, \tag{21}$$

*i.e., $\mathcal{L}_q(\eta_{ng}(t))$ converges to zero exponentially with rate 2.*

*Proof.* See Appendix C.3 □

Theorem 3 shows that gradient dynamics in the mixture family coordinates exhibit faster convergence rates than natural gradient dynamics, which, in turn, outperform gradient dynamics in the exponential family coordinates. On one hand, this supports the generally observed superiority of the natural gradient dynamics over gradient dynamics in the exponential family coordinates. On the other hand, it demonstrates that natural gradient dynamics are slower than gradient dynamics in the mixture family coordinates. Note that

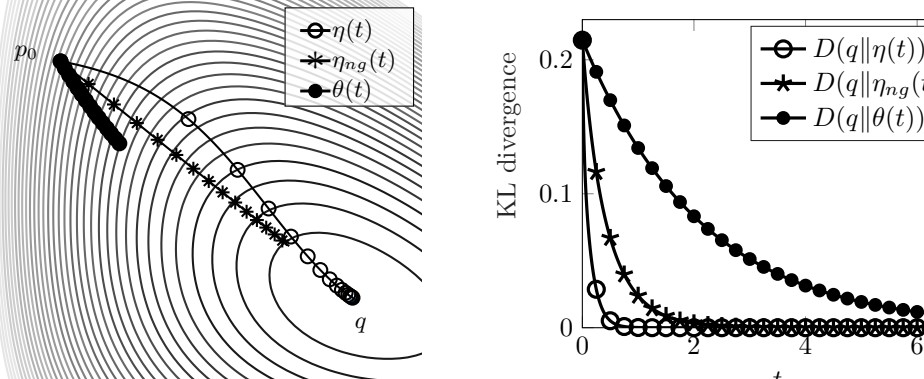

Figure 3: Left: Simulation trajectories of $\eta-$gradient flow described by equation 10, $\theta-$gradient flow described by equation 11 and the natural gradient flow described by equation 12 for $n = 2$ and $t \in [0, 1.5]$ superimposed on the level curves of KL divergence. The markers on the curves show equal time intervals for each curve. Right: KL divergence evaluated along the solutions plotted as a function of time $t$. The intervals between markers in the left figure are unrelated to the intervals between markers in the right figure.

although we choose to represent the natural gradient dynamics in the $\eta$ coordinates, the obtained convergence rate bound in equation 21 is independent of this choice. Furthermore, that the convergence rate bound for the $\eta$ coordinates established in equation 19 in Theorem 3 is asymptotic in nature and formally holds only after some time $T > 0$, whose value is not explicitly characterized. This is a common feature of asymptotic convergence rate analyses, where rate guarantees apply beyond some unknown $T$. Finally, as stated in the last sentence of item (i), Theorem 3, the result can be interpreted as describing local behavior: if the initial distribution lies sufficiently close to the target, the bound holds from the start (i.e., with $T = 0$). More precisely, a sufficient condition for the bound to hold with $T = 0$ is that $\nabla \mathcal{L}_q(\eta) \succ I$ for all $\eta$ belonging to the sublevel set $S_\eta := \{\eta \in \phi_m(S_n) | \mathcal{L}_q(\eta) \leq \mathcal{L}_q(\eta(0))\}$. Despite the absence of a global estimate on $T$, our empirical results (see Figure 4 for $n = 2$ and Figure 7 for $n = 10$) indicate that the ordering of convergence rates predicted by theory emerges early in the flow. This supports the practical relevance of the asymptotic comparison.

We now present numerical experiments with $n = 2$ to illustrate the theoretical results developed so far. Figure 3 (left) depicts the trajectories of the $\eta-$gradient flow described by equation 10, $\theta-$gradient flow described by equation 11 and the natural gradient flow described by equation 12 superimposed on the level curves of KL divergence. Although the natural gradient flow follows straight trajectories, it is slower than the $\eta-$gradient flow but faster than the $\theta-$gradient flow, as seen from the unit time markers along the curves. Figure 3 (right) confirms this by plotting the KL divergence over time along these trajectories.

To better highlight the convergence rates, Figure 4 (left) presents the KL divergence on a logarithmic scale, revealing exponential convergence. Best-fit linear curves are superimposed to estimate the slopes, which correspond to the convergence rates. The $\eta-$gradient flow exhibits the fastest convergence (slope $\approx 7$), the $\theta-$gradient flow is slowest (slope $\approx 0.475$), and the natural gradient flow lies in between (slope $\approx 2.04$), closely matching the theoretical prediction of rate 2. This experiment is repeated over 100 randomly chosen initial conditions and one randomly chosen target distribution, as shown in Figure 4 (right). The empirical convergence rates align well with the theoretical bounds from Theorem 3, confirming that $\eta-$gradient flows exceed rate 2, natural gradient flows converge at rate 2, and $\theta-$gradient flows fall below rate 2.

Finally, note that the natural gradient dynamics governed by equation 12 can be equivalently described as

$$\dot{\eta}_{ng}(t) = -\text{grad } \mathcal{L}_q(\eta_{ng}(t)) = \eta_q - \eta_{ng}(t) = -\nabla f_q(\eta_{ng}(t)),$$

where $f_q(\eta_{ng}(t)) = \frac{1}{2}\|\eta_q - \eta_{ng}(t)\|^2$. The natural gradient dynamics thus correspond to minimizing a convex quadratic function with identity Hessian.

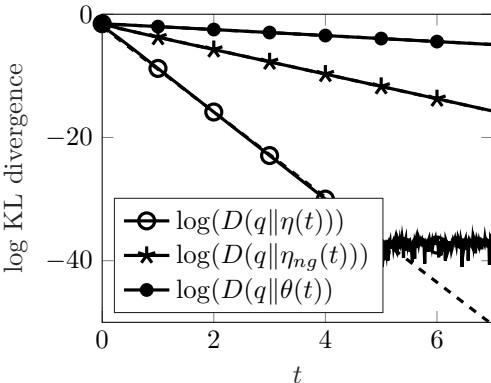
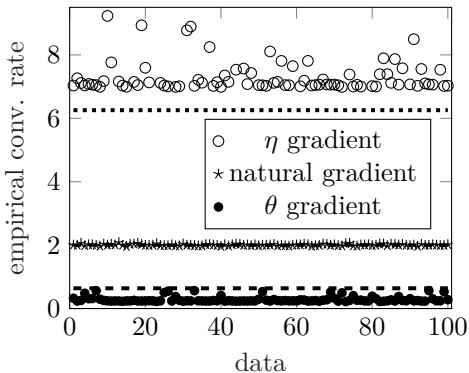

Figure 4: Left: KL divergence evaluated along the solutions to $\eta-$gradient flow described by equation 10, $\theta-$gradient flow described by equation 11 and the natural gradient flow described by equation 12 for $n = 2$ plotted on semi-log scale. The dashed lines show the best-fit linear function used to estimate the slope which gives the convergence rate. Right: Empirical convergence rates for 100 randomly chosen initial distributions and a randomly chosen target distribution for $n = 2$. The dotted line shows the theoretical lower bound for the convergence rate of $\eta-$gradient flows and the dashed line shows the theoretical upper bound for the convergence rate of $\theta-$gradient flows.

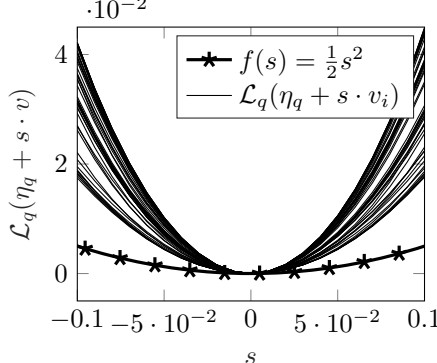
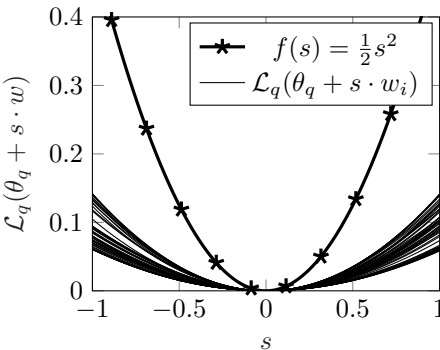

Figure 5: Left: Local sections of KL divergence around the minimizer $q$ plotted as $\mathcal{L}_q(\eta_q + s \cdot v_i)$, where all $v_i$ are unit norm vectors distributed evenly on the unit circle. A quadratic function $f(s) = \frac{1}{2}s^2$ is also shown for reference. Right: Local sections of KL divergence around the minimizer $q$ plotted as $\mathcal{L}_q(\theta_q + s \cdot w_i)$, where all $w_i$ are unit norm vectors distributed evenly on the unit circle. A quadratic function $f(s) = \frac{1}{2}s^2$ is also shown for reference.

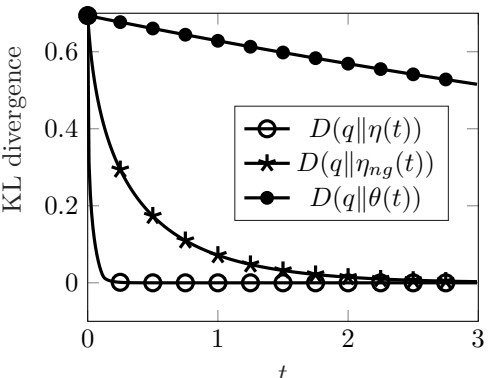 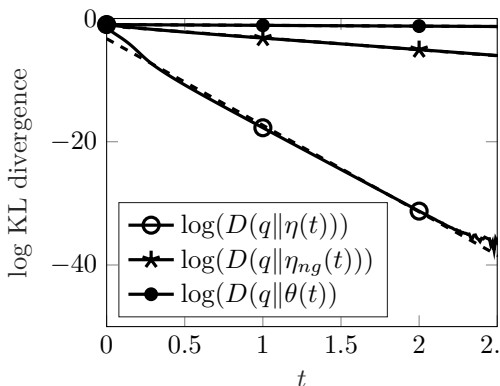

Figure 6: KL divergence evaluated along the solutions to $\eta-$gradient flow described by equation 10, $\theta-$gradient flow described by equation 11 and the natural gradient flow described by equation 12 for $n = 10$ plotted on a linear scale (left) and on a semi-log scale (right). The intervals between markers in the left figure are unrelated to the intervals between markers in the right figure. The dashed lines in the right figure show the best-fit linear function used to estimate the slope which gives the convergence rate.

Figure 5 offers insight into convergence behaviors by plotting local sections of the KL divergence near the optimum along different directions. This is achieved by plotting the functions $s \mapsto \mathcal{L}_q(\eta_q + s \cdot v_i)$ for different directions $v_i$ on the unit circle for the $\eta$ coordinates and by plotting the functions $s \mapsto \mathcal{L}_q(\theta_q + s \cdot w_i)$ for different directions $w_i$ on the unit circle for the $\theta$ coordinates. This illustrates the local curvature of the function around the optimum. Since the natural gradient dynamics correspond to minimizing a convex quadratic function with identity Hessian as discussed above, we overlay the plots with a quadratic function $f(s) = \frac{1}{2}s^2$ for reference. The plots reveal that the functions $s \mapsto \mathcal{L}_q(\eta_q + s \cdot v_i)$ exhibit higher curvature than the quadratic reference function $f$, while the functions $s \mapsto \mathcal{L}_q(\theta_q + s \cdot w_i)$ appear flatter. This provides the core intuition behind the fast convergence of the gradient flow in $\eta$ coordinates in comparison to gradient flow in the $\theta$ coordinates and shows why the natural gradient flow falls in between the two extremes.

To extend our numerical study to a higher-dimensional setting, we repeat the experiments from Figures 3 and 4 for the case $n = 10$. Figure 6 shows the KL divergence evaluated along gradient flow trajectories: the left panel shows the divergence on a linear scale, while the right panel uses a semi-logarithmic scale to highlight exponential convergence. As before, best-fit lines are overlaid to estimate the slopes, which correspond to the convergence rates. The $\eta-$gradient flow shows the fastest convergence (slope $\approx 16.239$), the $\theta-$gradient flow is the slowest (slope $\approx 0.099$), and the natural gradient flow lies in between (slope $\approx 1.974$), closely matching the theoretical rate of 2. For greater confidence, we repeat this experiment over 100 randomly chosen initial distributions and one randomly chosen target distribution; the results are shown in Figure 7. The empirical convergence rates align well with the theoretical predictions from Theorem 3, confirming that $\eta-$gradient flows exceed rate 2, natural gradient flows converge at rate 2, and $\theta-$gradient flows fall below rate 2. Moreover, based on these empirical studies, we also observe that the difference in convergence rates is more pronounced for $n = 10$ as compared to $n = 2$.

Finally, the analysis presented in this section raises an interesting question: Since the dual pairing between the coordinates $\eta$ and $\theta$ is preserved under appropriate affine transformations (as discussed in the following section and in Amari (2016), how do the convergence rates of the resulting dynamics change under such affine coordinate transformations? This question is addressed in the following subsection.

### 3.1 Convergence Rate Analysis Under Affine Coordinate Transformation

We first review the effect of an affine transformation of coordinates on the duality pairing between $\theta$ and $\eta$ (or equivalently between $\psi$ and $\varphi$). Since the convergence rate analysis from the previous section hinges on bounding the Hessian of the loss function, we investigate how the Hessian transforms under an affine change of coordinates. Consider new coordinates $\bar{\theta}$ that are related to the original $\theta$-coordinates via an affine

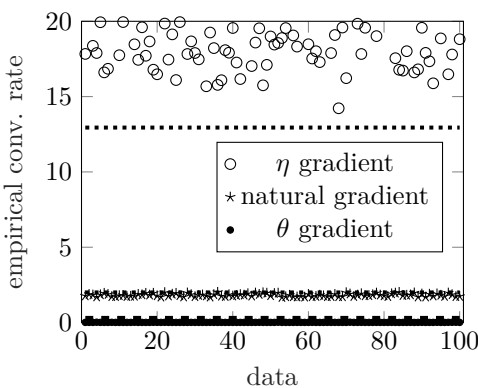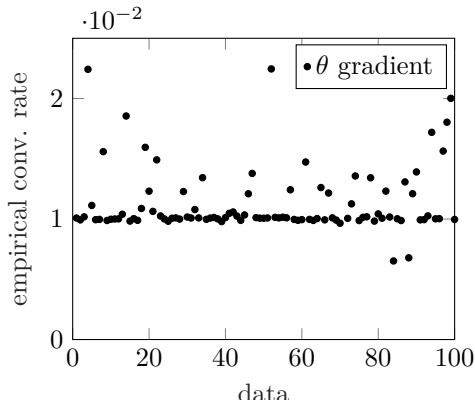

Figure 7: Left: Empirical convergence rates for 100 randomly chosen initial distributions and one randomly chosen target distribution with $n = 10$. The dotted line shows the theoretical lower bound (12.94) for the convergence rate of $\eta-$gradient flows and the dashed line shows the theoretical upper bound (0.31) for the convergence rate of $\theta-$gradient flows. Right: Same data, but with the y-axis restricted to highlight the variation in convergence rates of $\theta-$gradient flows.

transformation: $\theta = A\bar{\theta} + b$, where $A \in \mathbb{R}^{n \times n}$ is an invertible matrix and $b \in \mathbb{R}^n$ is an arbitrary vector. Let $\bar{\psi}$ be defined as $\bar{\psi}(\bar{\theta}) = \psi(A\bar{\theta} + b)$. A simple application of the chain rule shows $\nabla^2 \bar{\psi}(\bar{\theta}) = A^T \nabla^2 \psi(\theta) A$. Therefore, $\nabla^2 \bar{\psi}(\bar{\theta}) \succ 0$ if and only if $\nabla^2 \psi(\theta) \succ 0$, since $A$ is non-singular. This implies the strict convexity of $\bar{\psi}$, and it is possible to define its convex conjugate $\bar{\varphi}(\bar{\eta}) = \max_{\bar{\vartheta}} \left( \bar{\eta}^T \bar{\vartheta} - \bar{\psi}(\bar{\vartheta}) \right)$. Optimality condition on the maximizer $\bar{\vartheta}_{\text{opt}} = \bar{\theta}$ yields the relationship $\bar{\eta} = \nabla \bar{\psi}(\bar{\theta}) = A^T \nabla \psi(\theta) = A^T \eta$. It is straight-forward to verify that with these newly defined convex conjugate pairs of functions $\bar{\psi}$ and $\bar{\varphi}$ the Bregman divergence still gives the original KL divergence, i.e.,

$$D_{\bar{\psi}}(\bar{\theta}_p \parallel \bar{\theta}_q) = D_{\bar{\varphi}}(\bar{\eta}_q \parallel \bar{\eta}_p) = D_{\psi}(\theta_p \parallel \theta_q) = D_{\varphi}(\eta_q \parallel \eta_p) = D(q\|p).$$

The Hessians of the loss function when evaluated at the optimum, transform as follows:

$$\nabla^2 \mathcal{L}_q(\bar{\eta}_q) = \nabla^2 \bar{\varphi}(\bar{\eta}_q) = A^T \nabla^2 \varphi(\eta_q) A, \tag{22}$$

$$\nabla^2 \mathcal{L}_q(\bar{\theta}_q) = \nabla^2 \bar{\psi}(\bar{\theta}_q) = A^{-1} \nabla^2 \psi(\theta_q) A^{-T}. \tag{23}$$

This calculation immediately gives us the following theorem.

**Theorem 4** (Dual coordinates with identity Hessian). *Let $c$ be a positive constant, $q \in S_n$ be the target distribution and $p_0 \in S_n$ be the initial distribution. There exists a pair of convex conjugate functions $\bar{\psi}$ and $\bar{\varphi}$ inducing the pair of dual coordinates $\bar{\eta}$ and $\bar{\theta}$ for $S_n$ with coordinate maps $\bar{\phi}_m$ and $\bar{\phi}_e$ such that*

$$\nabla^2 \mathcal{L}_q(\bar{\eta}_q) = \left[ \nabla^2 \mathcal{L}_q(\bar{\theta}_q) \right]^{-1} = c \cdot I. \tag{24}$$

*Consider the gradient flow dynamics:*

$$\dot{\bar{\eta}}(t) = -\nabla \mathcal{L}_q(\bar{\eta}(t)), \qquad \bar{\eta}(0) = \bar{\eta}_{p_0},$$

$$\dot{\bar{\theta}}(t) = -\nabla \mathcal{L}_q(\bar{\theta}(t)), \qquad \bar{\theta}(0) = \bar{\theta}_{p_0}.$$

*Then for any $\varepsilon > 0$, there exist positive constants $c_1$, $c_2$, $c_3$, $c_4$ and $T$ such that for all $t \geq T$,*

$$c_1 e^{-2(c+\varepsilon)t} \leq \mathcal{L}_q(\bar{\eta}(t)) \leq c_2 e^{-2(c-\varepsilon)t}, \tag{25}$$

$$c_3 e^{-2(\frac{1}{c}+\varepsilon)t} \leq \mathcal{L}_q(\bar{\theta}(t)) \leq c_4 e^{-2(\frac{1}{c}-\varepsilon)t}, \tag{26}$$

*i.e., $\mathcal{L}_q(\bar{\eta}(t))$ and $\mathcal{L}_q(\bar{\theta}(t))$ converge exponentially with rate $2c$ and $\frac{2}{c}$, respectively.*

*Proof.* See Appendix C.4 □

Note that by plugging $c = 1$ in Theorem 4, we see that there exists a dual pair of coordinates that achieves the convergence rate of the natural gradient dynamics. However, the affine transformation that leads to this convergence rate depends on the target distribution $q$ and thus cannot be known in advance. Furthermore, Theorem 4 illustrates that the convergence rate of gradient flows in the transformed coordinates can be made arbitrarily small or arbitrarily large by scaling the coordinates. In contrast, the natural gradient flow is independent of the choice of coordinates, and therefore, has a coordinate independent convergence rate.

The superiority of the natural gradient method in terms of convergence rates is not immediately clear from the continuous-time analysis presented so far. In order to facilitate a meaningful discussion of convergence rates in continuous time, Muehlebach & Jordan (2020) propose a particular time-normalization approach that can be deployed in our setting. Alternatively, a more direct comparison between the Euclidean gradient and the natural gradient can be made by studying discrete-time gradient descent iterations. To elaborate this, we turn our attention to the discrete-time setting in the next section, where the advantages of the natural gradient method become evident.

## 4 Convergence Analysis in Discrete Time

For a given target distribution $q \in S_n$ and an initial distribution $p_0 \in S_n$, the discrete-time gradient dynamics are given by

$$\eta(k+1) = \eta(k) - \alpha_\eta \cdot \nabla \mathcal{L}_q(\eta(k)) = \eta(k) + \alpha_\eta \nabla^2 \varphi(\eta(k))(\eta_q - \eta(k)), \qquad \eta(0) = \eta_{p_0}, \qquad (27)$$

$$\theta(k+1) = \theta(k) - \alpha_\theta \cdot \nabla \mathcal{L}_q(\theta(k))) = \theta(k) - \alpha_\theta \nabla \psi(\theta(k)) + \alpha_\theta \nabla \psi(\theta_q), \qquad \theta(0) = \theta_{p_0}, \qquad (28)$$

$$\eta_{ng}(k+1) = \eta_{ng}(k) - \alpha_{ng} \cdot \text{grad } \mathcal{L}_q(\eta_{ng}(k)) = \eta_{ng}(k) - \alpha_{ng}\left(\eta_{ng}(k) - \eta_q\right), \quad \eta_{ng}(0) = \eta_{p_0}, \qquad (29)$$

where $\alpha_\eta$, $\alpha_\theta$ and $\alpha_{ng}$ are the learning rates. Unlike in the continuous-time setting, the choice of coordinates used to represent the natural gradient dynamics in discrete time influences the analysis of convergence rates, primarily due to the presence of the learning rate $\alpha$ in the update equations (see Martens (2020); Song et al. (2018)). In what follows, we restrict the analysis to the representation of the natural gradient in the $\eta$ coordinates.

To simplify the analysis, let us linearize these dynamics around the equilibrium points and examine the local convergence rates of the linearized dynamics. Owing to the already linear natural gradient dynamics, these do not need to be linearized. These linearized dynamics are given by

$$\eta(k+1) = \left(I - \alpha_\eta \nabla^2 \varphi(\eta_q)\right)\eta(k) + \alpha_\eta \nabla^2 \varphi(\eta_q)\eta_q, \qquad \eta(0) = \eta_{p_0}, \qquad (30)$$

$$\theta(k+1) = \left(I - \alpha_\theta \nabla^2 \psi(\theta_q)\right)\theta(k) + \alpha_\theta \nabla^2 \psi(\theta_q)\theta_q, \qquad \theta(0) = \theta_{p_0}, \qquad (31)$$

$$\eta_{ng}(k+1) = (1 - \alpha_{ng})\eta_{ng}(k) + \alpha_{ng} \cdot \eta_q, \qquad \eta_{ng}(0) = \eta_{p_0}. \qquad (32)$$

It turns out that the discrete-time natural gradient dynamics in the $\theta$ coordinates, when linearized about the equilibrium $\theta_q$, leads to update equations that are identical to equation 32. This is elaborated in Appendix B. Furthermore, also note that the update equation 32 is invariant to any affine transformation of the coordinates. Therefore, the update equation 32 represents local linearized dynamics for all dual pairs of coordinates.

The dynamics described by equation 30, equation 31 and equation 32 can be written in the general form

$$x(k+1) = (I - \alpha Q)\,x(k) + \alpha Q x^*$$

where $Q$ is a symmetric positive definite matrix. Notice that these dynamics result from gradient descent iterations when optimizing the convex quadratic function $f(x) = \frac{1}{2}(x - x^*)^T Q(x - x^*)$. In this discrete-time setting, we say that a sequence $f(x(k))$ converges exponentially to $f(x_*)$ with rate $\rho \in [0, 1)$ if there exists a positive constant $c$ and an integer $k_0$ such that $|f(x(k)) - f(x_*)| \leq c\rho^k$ holds for all $k \geq k_0$. A smaller value of $\rho$ corresponds to faster convergence. The convergence rates of gradient descent algorithms for minimizing convex quadratic functions have been extensively studied. For example, the following result from Nesterov (2018) shows that the condition number of $Q$ determines the convergence rates.

**Theorem 5** (Nesterov (2018)). *Let $f(x) = \frac{1}{2}(x - x^*)^\top Q(x - x^*)$ with $Q \succ 0$, and let $\kappa$ denote the condition number of $Q$. Consider the gradient descent iterations $x(k+1) = x(k) - \alpha \nabla f(x(k))$. Then:*

*(i) $f(x(k))$ converges to zero at rate $\left(1 - \frac{1}{\kappa}\right)^2$ when $\alpha = \frac{1}{\lambda_{max}(Q)}$ (standard choice).*

*(ii) $f(x(k))$ converges to zero at rate $\left(1 - \frac{2}{\kappa+1}\right)^2$ when $\alpha = \frac{2}{\lambda_{min}(Q) + \lambda_{max}(Q)}$ (optimal choice).*

Note that the gradient descent dynamics in $\eta$ and $\theta$ coordinates correspond to setting $Q = \nabla^2 \varphi(\eta_q)$ and $Q = \nabla^2 \psi(\theta_q)$, respectively. By directly applying these results to the discrete-time gradient descent dynamics described by equation 30 and equation 31, we observe that poorer conditioning of $Q$ leads to a larger convergence rate $\rho$, and thus slower convergence. The condition numbers associated with the gradient descent dynamics in the $\eta$ and $\theta$ coordinates can be bounded away from 1 as stated in Lemma 6.

**Lemma 6** (Bounds on the condition number of the Hessian). *Let $q \in S_n$. Then,*

$$1 < \kappa_q \leq \text{cond}(\nabla^2 \mathcal{L}_q(\eta_q)) = \text{cond}(\nabla^2 \mathcal{L}_q(\theta_q)), \tag{33}$$

*where $\kappa_q = \frac{\eta_{\min,2}}{\eta_{\min}}$ with $\eta_{\min}$ and $\eta_{\min,2}$ being the smallest and the second-smallest element of $\{[\eta_q]_1, \cdots, [\eta_q]_n\}$, respectively.*

*Proof.* See Appendix C.5 □

The natural gradient dynamics correspond to setting $Q = I$ which yields optimal conditioning. Observe that the discrete-time natural gradient descent dynamics achieve a convergence rate of $|1 - \alpha|$ for $\alpha \in (0, 2)$. Furthermore, it achieves an optimal convergence rate of 0, i.e., convergence in a single step for the optimal learning rate $\alpha = 1$. Note, however, that since this analysis pertains to the linearized system, the actual natural gradient descent does not converge in single step. Finally, with the goal of studying the properties of the stochastic gradient descent (SGD), we examine the robustness of these dynamics to imperfect gradient measurements. Although the noise models studied next do not exactly model the stochastic behavior of the SGD, they take us a step closer to it and provide valuable insight. Furthermore, practical implementations of the natural gradient method involve approximating the Fisher information matrix by an empirical version of it Martens (2020). This can also be captured to some degree by the noise models studied next. Specifically, we study two noise models motivated by (Polyak, 1987, Chapter 4): relative deterministic noise (multiplicative) and absolute random noise (additive). These and other similar noise models have been studied in the optimization literature and they evidently show that the condition number plays a central role in these analyses (see Guille-Escuret et al. (2021); Lessard et al. (2016); Van Scoy & Lessard (2021)).

### 4.1 Robustness Analysis with Relative Deterministic Noise

Let us first consider the relative deterministic noise (multiplicative) model which replaces the gradient vector $v$ by $(I + \Delta(k))v$ where $\Delta(k) \in \mathbb{R}^{n \times n}$ captures the noise at time instant $k$. This leads to dynamics

$$\eta(k+1) = \eta(k) - \alpha_\eta \cdot (I + \Delta(k)) \nabla^2 \varphi(\eta_q)(\eta(k) - \eta_q), \qquad \eta(0) = \eta_{p_0}, \tag{34}$$

$$\theta(k+1) = \theta(k) - \alpha_\theta \cdot (I + \Delta(k)) \nabla^2 \psi(\theta_q)(\theta(k) - \theta_q), \qquad \theta(0) = \theta_{p_0}, \tag{35}$$

$$\eta_{ng}(k+1) = \eta_{ng}(k) - \alpha_{ng} \cdot (I + \Delta(k))(\eta_{ng}(k) - \eta_q), \qquad \eta_{ng}(0) = \eta_{p_0}, \tag{36}$$

where the learning rates $\alpha_\eta$, $\alpha_\theta$ and $\alpha_{ng}$ are chosen optimally assuming the noise-free conditions ($\Delta \equiv 0$).

**Theorem 7** (Robust stability under relative deterministic noise). *Consider a target distribution $q \in S_n$, an initial distribution $p \in S_n$ and the discrete-time dynamics described by equation 34, equation 35 and equation 36, respectively, where the learning rates $\alpha_\eta$, $\alpha_\theta$ and $\alpha_{ng}$ are chosen optimally for each case assuming the absence of noise ($\Delta \equiv 0$). Let $\kappa = \text{cond}(\nabla^2 \varphi(\eta_q)) = \text{cond}(\nabla^2 \psi(\theta_q))$. Then the following statements hold:*

(i) *If the sequence of perturbations $\Delta(k)$ is such that for some $\varepsilon > 0$, $\|\Delta(k)\|_2 < 1 - \varepsilon$ for all $k \geq 0$, then the natural gradient dynamics described by equation 36 are stable, i.e., $\lim_{k \to \infty} \|\eta_{ng}(k) - \eta_q\| = 0$.*

(ii) *There exist time-invariant perturbations $\Delta_\eta$ and $\Delta_\theta$ with $\|\Delta_\eta\|_2 = \|\Delta_\theta\|_2 = \frac{1}{\kappa}$ that destabilize the gradient descent dynamics described by equation 34 and equation 35, respectively.*

*Proof.* See Appendix C.6 □

The above result shows that the natural gradient dynamics exhibit a larger robustness margin in comparison to the robustness margin of the $\eta$ and $\theta$ gradient dynamics which depend on the condition number $\kappa$. This shows that the superiority of the natural gradient dynamics can be again attributed to the optimal conditioning ($\kappa = 1$). Also note that the above noise model includes time-varying perturbations to the learning rate and shows that the natural gradient dynamics tolerate a much higher deviation from the optimal learning rate. Furthermore, note that statement (i) of the above theorem proves convergence for the situation where the inverse of the Fisher information matrix $G^{-1}$ is replaced by $(I + \Delta(k))G^{-1}$ with $\|\Delta(k)\|_2 < 1 - \varepsilon$ for all $k \geq 0$. This result thus also makes progress towards the more practical implementations of the natural gradient involving an empirical version of the Fisher information matrix Martens (2020).

## 4.2 Robustness Analysis with Absolute Random Noise

Now let us now consider the absolute random noise (additive) model which perturbs the original dynamics by adding an independent and identically distributed noise signal $\delta(k)$ for $k \in \{0, 1, \cdots\}$. This leads to dynamics

$$\eta(k+1) = \eta(k) - \alpha_\eta \left(\nabla^2 \varphi(\eta_q)(\eta(k) - \eta_q)\right) + \delta(k), \qquad \eta(0) = \eta_{p_0} \tag{37}$$

$$\theta(k+1) = \theta(k) - \alpha_\theta \left(\nabla^2 \psi(\theta_q)(\theta(k) - \theta_q)\right) + \delta(k), \qquad \theta(0) = \theta_{p_0}, \tag{38}$$

$$\eta_{ng}(k+1) = \eta_{ng}(k) - \alpha_{ng} \left(\eta_{ng}(k) - \eta_q\right) + \delta(k), \qquad \eta_{ng}(0) = \eta_{p_0}, \tag{39}$$

where learning rates $\alpha_\eta$, $\alpha_\theta$ and $\alpha_{ng}$ are chosen optimally for each case assuming the absence of noise ($\delta(k) \equiv 0$). Furthermore, assume that $\delta(k)$ is an independent and identically distributed stochastic process satisfying $\mathbb{E}[\delta(k)] = 0$ and $\mathbb{E}[\delta(k)\delta(k)^T] = I$ for all $k \geq 0$.

**Theorem 8** (Robustness against additive noise)**.** *Consider a target distribution $q \in S_n$, an initial distribution $p_0 \in S_n$ and the discrete-time dynamics described by equation 37, equation 38 and equation 39, respectively, where the learning rates $\alpha_\eta$, $\alpha_\theta$ and $\alpha_{ng}$ are chosen optimally for each case assuming the absence of noise ($\delta(k) \equiv 0$). Then*

(i) $\lim_{k \to \infty} \mathbb{E}[(\eta(k) - \eta_q)(\eta(k) - \eta_q)^T] = \Sigma_\eta \preceq \frac{(\kappa+1)^2}{4\kappa} I$,

(ii) $\lim_{k \to \infty} \mathbb{E}[(\theta(k) - \theta_q)(\theta(k) - \theta_q)^T] = \Sigma_\theta \preceq \frac{(\kappa+1)^2}{4\kappa} I$,

(iii) $\mathbb{E}[(\eta_{ng}(k) - \eta_q)(\eta_{ng}(k) - \eta_q)^T] = I$ for all $k \geq 0$.

*The upperbound in (i) and (ii) is tight, i.e., $\Sigma_\eta$ and $\Sigma_\theta$ have eigenvalues equal to $\frac{(\kappa+1)^2}{4\kappa}$. Furthermore, for $n = 2$, we get equality in (i) and (ii).*

*Proof.* See Appendix C.7 □

The above result explores the effect of adding an independent and identically distributed (i.i.d.) noise signal at every iteration of the dynamics. It establishes that the largest eigenvalues of the steady-state error covariances are given by $\frac{(\kappa+1)^2}{4\kappa} I$ for the $\eta$ and $\theta$ gradient dynamics whereas the error covariance with the natural gradient dynamics equals the noise covariance which corresponds to optimal conditioning ($\kappa = 1$).

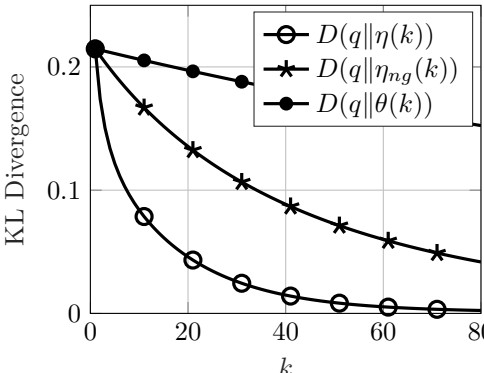 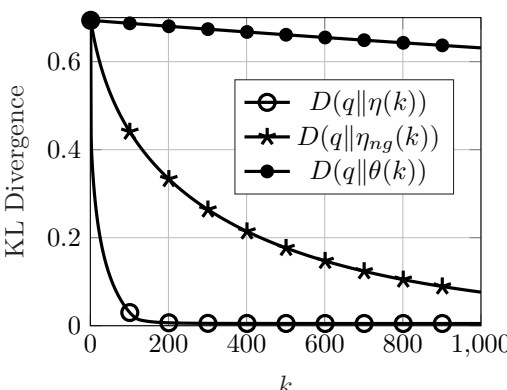

Figure 8: KL divergence evaluated along optimization trajectories generated by discrete-time gradient descent dynamics described by equation 40, equation 41 and equation 42. All methods use the same and sufficiently small learning rate. The left panel shows results for $n = 2$ with $\alpha_\eta = \alpha_\theta = \alpha_{ng} = 0.01$; the right panel shows $n = 10$ and $\alpha_\eta = \alpha_\theta = \alpha_{ng} = 0.001$. Convergence behavior is consistent with its continuous-time counterpart.

## 5 Numerical Experiments with Empirical KL Divergence

The gradient expressions analyzed in previous sections involve the target distribution $q$, which is typically unknown in practical machine learning applications. In reality, one only has access to a data sequence $\mathcal{D} = (x_1, x_2, \cdots, x_N)$ consisting of samples drawn i.i.d. with respect to $q$. This section aims to bridge the gap between our theoretical analysis and practical implementations by investigating empirical versions of the KL divergence and their associated optimization dynamics. Recall that the KL divergence can be decomposed as

$$\mathcal{L}_q(p) = \sum_{i=1}^{n+1} q_i \log\left(\frac{q_i}{p_i}\right) = H(q) - \sum_{i=1}^{n+1} q_i \log(p_i),$$

where $H(q)$ is the (negative) Shannon entropy of $q$, independent of $p$, and the second term is the cross-entropy, equal to $-\mathbb{E}_{x \sim q}[\log(p_x)]$. As a result, minimizing $\mathcal{L}_q(p)$ is equivalent to minimizing the cross-entropy term, since the entropy term is constant with respect to $p$. Given a data sequence $\mathcal{D}$ sampled with respect to $q$, we can estimate this expectation using the empirical mean

$$-\frac{1}{N} \sum_{j=1}^{N} \log(p_{x_j}) = -\sum_{i=1}^{n+1} \hat{q}_i \log(p_i),$$

where $\hat{q}_i$ is the relative frequency of outcome $i$ in $\mathcal{D}$. This defines the empirical target distribution $\hat{q} = (\hat{q}_1, \cdots, \hat{q}_{n+1}) \in S_n$. This motivates the definition of the empirical KL divergence, defined for a given data sequence $\mathcal{D}$, as $\hat{\mathcal{L}}_{\mathcal{D}}(p) = \mathcal{L}_{\hat{q}}(p)$. Since this differs from the empirical cross-entropy only by a constant (independent of $p$), minimizing $\hat{\mathcal{L}}_{\mathcal{D}}(p)$ is equivalent to minimizing the empirical estimate of $\mathcal{L}_q(p)$. Therefore, our continuous-time analysis in Section 3 applies directly in this empirical setting by substituting $q$ with $\hat{q}$, assuming $\hat{q}$ lies in the interior of the simplex $S_n$, which typically holds for large enough data sequences. As $N \to \infty$, we have that $\hat{q} \to q$, ensuring that the empirical KL divergence converges to the true KL divergence. This provides a direct connection between our theoretical results and their applicability in data-driven settings.

In what follows, we outline the setting described above more explicitly. Consider an arbitrary target distribution $q$ and generate the data sequence $\mathcal{D}$ of independent and identically distributed samples drawn with respect to $q$. Analogous to the discrete-time updates given in equations equation 27, equation 28, and

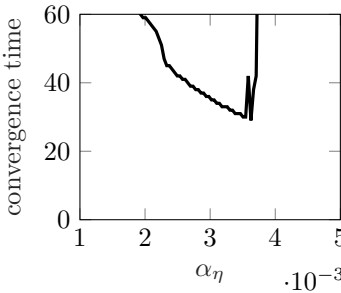 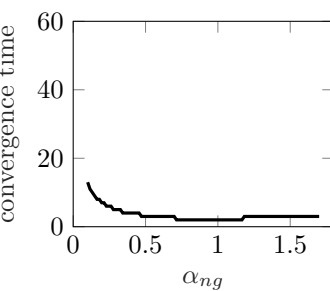 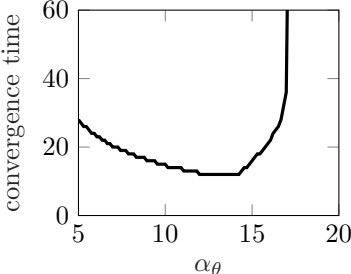

Figure 9: Convergence time as a function of learning rate for discrete-time gradient descent dynamics governed by equation 40 (left), equation 41 (right) and equation 42 (center). Results correspond to dimension $n = 10$. For each learning rate, convergence time is defined as the maximum number of iterations (across 100 random initializations) required to reach a fixed tolerance near the optimum. Natural gradient descent achieves the fastest convergence when appropriately tuned, as evidenced by comparing the lowest convergence times in each plot. These results are summarized in Table 1

equation 29, consider the empirical gradient descent dynamics described by

$$\eta(k+1) = \eta(k) - \alpha_\eta \cdot \nabla \hat{\mathcal{L}}_{\mathcal{D}}(\eta(k)), \qquad \eta(0) = \eta_{p_0}, \tag{40}$$

$$\theta(k+1) = \theta(k) - \alpha_\theta \cdot \nabla \hat{\mathcal{L}}_{\mathcal{D}}(\theta(k))), \qquad \theta(0) = \theta_{p_0}, \tag{41}$$

$$\eta_{ng}(k+1) = \eta_{ng}(k) - \alpha_{ng} \cdot \operatorname{grad} \hat{\mathcal{L}}_{\mathcal{D}}(\eta_{ng}(k)), \quad \eta_{ng}(0) = \eta_{p_0}. \tag{42}$$

For sufficiently small and equal learning rates $\alpha_\eta = \alpha_\theta = \alpha_{ng}$, the discrete-time dynamics approximate the continuous-time behavior. This is illustrated in Figure 8, where we set $\alpha_\eta = \alpha_\theta = \alpha_{ng} = 0.01$ for $n = 2$ (left) and $\alpha_\eta = \alpha_\theta = \alpha_{ng} = 0.001$ for $n = 10$ (right). The sandwiching property derived in the continuous-time setting (see Theorem 3) thus extends, as expected, to the discrete-time setting when the learning rates are chosen to be equal across the different methods and sufficiently small.

Drawing motivation from Theorem 5, which analyzed convergence rates under different choices of learning rates, we next explore how varying the learning rates individually for each method influences empirical convergence properties. specifically, we sample 100 random initializations and, for each algorithm and each learning rate, run simulations from these 100 initializations. We then measure the convergence time defined as the maximum umber of iterations (across the 100 initializations) required to reach a fixed tolerance from the optimum. Figure 9 plots convergence time as a function of learning rate for discrete-time gradient descent dynamics governed by equation 40, equation 41 and equation 42. Results correspond to dimension $n = 10$. The optimal learning rates and convergence times are shown in Table 1.

Table 1: Optimal Learning Rates and Convergence Times from Figure 9

|  | optimal learning rate | optimal convergence time |
|---|---|---|
| $\eta$ coordinates | $\alpha_\eta = 0.0036$ | $k = 29$ |
| natural gradient | $\alpha_{ng} \in [0.7141, 1.16]$ | $k = 2$ |
| $\theta$ coordinates | $\alpha_\theta \in [11.96, 14.24]$ | $k = 12$ |

With optimally tuned learning rates, natural gradient descent consistently converges faster than standard gradient descent in either coordinate system.

## 5.1 Numerical Experiments with Stochastic Gradient Descent

Recall that the key motivation behind Theorems 7 and 8 is to go beyond gradient descent potentially taking a step towards better understanding the behavior of stochastic gradient descent (SGD). To this end, we now extend our empirical study to the SGD setting, where at each iteration $k$, a mini-batch $\mathcal{D}_k \subset \mathcal{D}$ is

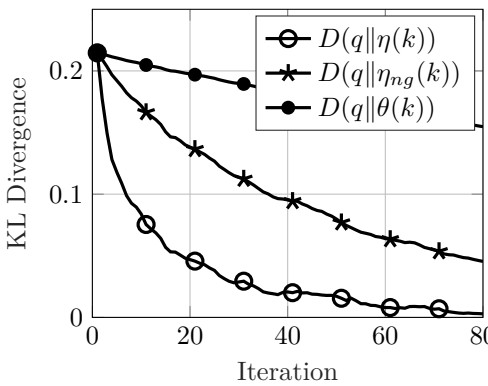 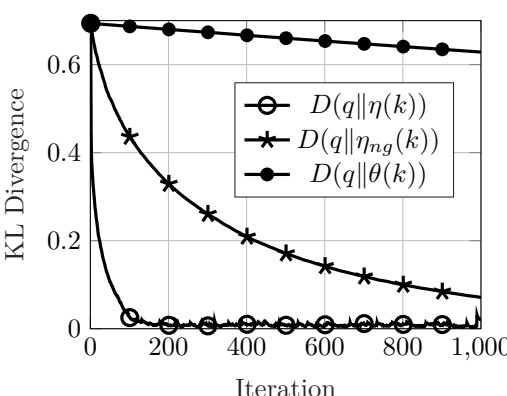

Figure 10: KL divergence evaluated along optimization trajectories generated by discrete-time gradient descent dynamics described by equation 43, equation 44 and equation 45. All methods use the same and sufficiently small learning rate. The left panel shows results for $n = 2$ with $\alpha_\eta = \alpha_\theta = \alpha_{ng} = 0.01$; the right panel shows $n = 10$ and $\alpha_\eta = \alpha_\theta = \alpha_{ng} = 0.001$. Convergence behavior is consistent with its continuous-time counterpart.

drawn uniformly at random, and gradients are computed with respect to the empirical loss $\hat{\mathcal{L}}_{\mathcal{D}_k}$. To satisfy standard conditions for the almost sure convergence of SGD, we adopt a time-varying learning rate of the form $\alpha \cdot \left(\frac{a}{k+a}\right)$, where the parameter $a$ is chosen large enough to delay the onset of decay until sufficiently large $k$ and is kept equal across all the different algorithms. The resulting dynamic equations are then described by

$$\eta(k + 1) = \eta(k) - \alpha_\eta \cdot \left(\frac{a}{k + a}\right) \cdot \nabla \hat{\mathcal{L}}_{\mathcal{D}_k}(\eta(k)), \qquad \eta(0) = \eta_{p_0}, \tag{43}$$

$$\theta(k + 1) = \theta(k) - \alpha_\theta \cdot \left(\frac{a}{k + a}\right) \cdot \nabla \hat{\mathcal{L}}_{\mathcal{D}_k}(\theta(k))), \qquad \theta(0) = \theta_{p_0}, \tag{44}$$

$$\eta_{ng}(k + 1) = \eta_{ng}(k) - \alpha_{ng} \cdot \left(\frac{a}{k + a}\right) \cdot \text{grad } \hat{\mathcal{L}}_{\mathcal{D}_k}(\eta_{ng}(k)), \quad \eta_{ng}(0) = \eta_{p_0}. \tag{45}$$

Analogous to the full-batch setting, Figure 10 shows that for sufficiently small and equal learning rates $\alpha_\eta = \alpha_\theta = \alpha_{ng}$, the sandwiching property derived in the continuous-time setting (see Theorem 3) extends to the setting of stochastic gradient descent. We next plot convergence times as a function of learning rate in Figure 11 and summarize the optimal values in Table 2. Once again, stochastic natural gradient descent

Table 2: Optimal Learning Rates and Convergence Times from Figure 11

|  | **optimal learning rate** | **optimal convergence time** |
|---|---|---|
| $\eta$ coordinates | $\alpha_\eta = 0.0025$ | $k = 41$ |
| natural gradient | $\alpha_{ng} \in [0.541, 0.695]$ | $k = 3$ |
| $\theta$ coordinates | $\alpha_\theta \in 6.93$ | $k = 20$ |

outperforms its Euclidean counterparts when the learning rate is optimally tuned. Figure 12 provides example trajectories illustrating the convergence behavior under both full-batch and stochastic updates with optimally chosen learning rates. In contrast to sandwiching property observed in Figure 8 and Figure 10, it can be seen in Figure 12 that the natural gradient trajectories converge faster than their Euclidean counterparts in the full-batch as well as stochastic settings.

## 5.2 Summary

These empirical studies validate that our theoretical insights apply to practical scenarios involving sampled data from the target distribution. In both full-batch and stochastic optimization settings, natural gradient

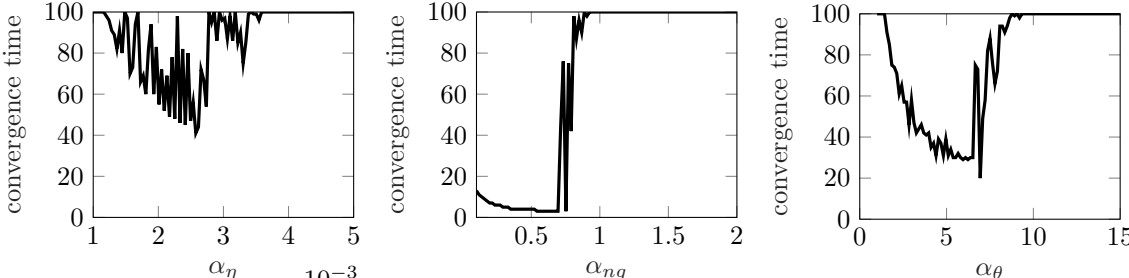

Figure 11: Convergence time as a function of learning rate for discrete-time stochastic gradient descent governed by equation 43 (left), equation 44 (right) and equation 45 (center). Results correspond to dimension $n = 10$. For each learning rate, convergence time is defined as the maximum number of iterations (across 100 random initializations) required to reach a fixed tolerance near the optimum. Natural gradient descent achieves the fastest convergence when appropriately tuned, as evidenced by comparing the lowest convergence times in each plot. These results are summarized in Table 2

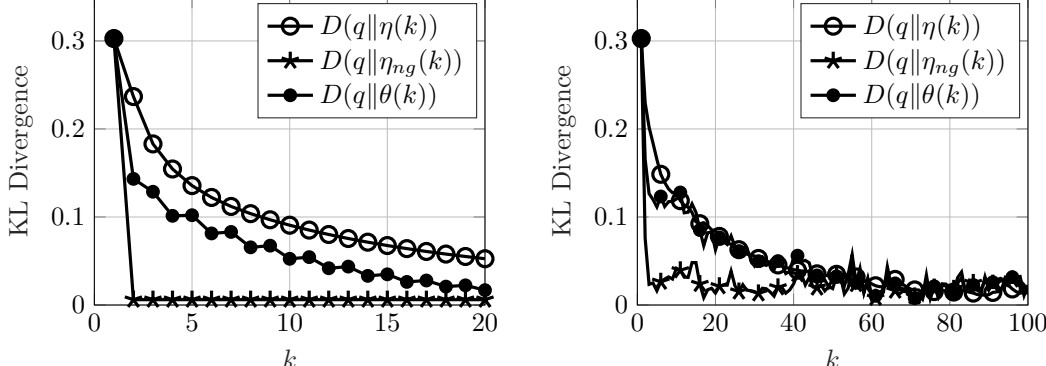

Figure 12: KL divergence evaluated along the trajectories generated by equation 40, equation 41 and equation 42 (left) corresponding the full-batch setting and by equation 43, equation 44 and equation 45 (right) corresponding to the stochastic setting. Results correspond to dimension $n = 10$ and the learning rates are set to their optimal values obtained from Figure 9 for the left panel and from Figure 11 for the right panel. Natural gradient descent exhibits faster convergence across both settings with appropriately tuned learning rates.

descent (NGD) consistently outperforms standard gradient descent (GD) under optimally tuned learning rates. This corresponds to Theorems 5, 7, and 8. In contrast, when learning rates are chosen to be equal for all methods and sufficiently small, the sandwiching property established in Theorem 3 for continuous-time dynamics holds also in the discrete-time setting.

Table 3 summarizes the correspondence between theoretical results and empirical findings. These empirical observations thus closely align with the theoretical results developed in the preceding sections, reinforcing the practical relevance of our analysis.

Table 3: Summary of Theoretical and Empirical Results

| Setting | Sandwiching property ($\alpha_\eta = \alpha_{ng} = \alpha_\theta$ sufficiently small) | NGD outperforms GD ($\alpha_\eta, \alpha_{ng}, \alpha_\theta$ individually optimized) |
|---|---|---|
| Theory | Theorem 3 | Theorems 5, 7, 8 |
| Empirical studies (GD) | Figure 8 | Table 1, Figure 12 (left) |
| Empirical studies (SGD) | Figure 10 | Table 2, Figure 12 (right) |

## 6  Conclusions and Outlook

In this work, we revisited the convergence properties of natural gradient flows in comparison to their Euclidean counterparts, focusing on the minimization of the KL divergence over discrete probability distributions. Our analysis revealed a more nuanced picture than the commonly held belief in the universal superiority of natural gradient methods. In the continuous-time setting, while the natural gradient flow indeed outperforms the Euclidean gradient flow in the $\theta$ coordinates, consistent with traditional expectations, we showed that it converges more slowly than the $\eta$-gradient flow, despite following straight-line trajectories in these coordinates. This demonstrates that the commonly observed rapid convergence of natural gradient flow cannot be simplistically attributed to the straightness of its trajectories. Our discrete-time analysis of gradient descent dynamics further clarified that the fundamental reason behind the superiority of natural gradient methods lies in their optimal conditioning: natural gradient updates effectively minimize an optimally conditioned loss landscape, leading to consistently better performance compared to their Euclidean counterparts. Furthermore, our empirical results reinforce the theoretical results by demonstrating that natural gradient descent (NGD) maintains its convergence advantages over standard gradient descent (GD) even in practical, sample-based settings. In both full-batch and stochastic optimization, NGD exhibits faster convergence when learning rates are optimally tuned, and satisfies the theoretical sandwiching property under sufficiently small and equal learning rates. These findings highlight the practical relevance of the theoretical analyses and support the use of NGD in real-world applications where access to the full distribution is limited to finite samples. Overall, our findings refine the understanding of natural gradient methods and highlight the subtle, yet important, nuances that govern their behavior.

The theoretical results presented in this paper primarily concern exact natural gradient, with robustness analyses taking initial steps toward incorporating noise in gradient measurements and system dynamics. A natural next step is to extend this analysis to practical settings where the Fisher information matrix must be approximated. For instance, Theorem 7 already accommodates structured perturbations of the form $(I + \Delta(k))G^{-1}$, where $G$ is the Fisher information matrix, provided $\|\Delta(k)\|_2 < 1 - \varepsilon$ for all $k \geq 0$, ensuring the stability of the resulting updates. This suggests that if approximations such as K-FAC can be modeled as structured perturbations of $G^{-1}$ satisfying the above condition, then it may be possible to establish convergence guarantees for the resulting approximate natural gradient algorithms. A systematic exploration of the trade-off between the quality of the approximation of the Fisher matrix and the resulting convergence guarantees would be an important direction for bridging theory and practice.

While we focused on the probability simplex equipped with dual coordinate systems, an important extension would be to general dually flat statistical manifolds (Amari & Nagaoka (2000); Ay et al. (2017)), where similar tools may be employed to study optimization dynamics in broader settings. Such manifolds are induced by general Bregman divergences going beyond the setting of a KL divergence. Notably, the identity $\nabla^2 \varphi(\eta) = [\nabla^2 \psi(\theta)]^{-1}$ holds for any pair of dual Bregman divergences induced by strictly convex functions $\varphi$

and $\psi$, respectively Amari & Nagaoka (2000). So it is plausible that one can derive an analogue of Lemma 2 for this general setting provided $\nabla^2 \varphi(\eta) \succ I$ holds at the optimum. One can then proceed to derive the "sandwich" ordering as in Theorem 3.

Additionally, our framework may be extended to richer families of probability distributions, such as general exponential families derived from Boltzmann machines without hidden units, and more intricate mixtures of exponential families associated with Boltzmann machines with hidden variables (Amari et al. (1992)). These models exhibit more complex geometries that may reveal deeper interactions between parametrizations and optimization dynamics.

While our analysis in Section 4 relies on a linearization of the gradient flows around the optimal solution (i.e., the target distribution $q$), we acknowledge that extending these results to the fully nonlinear setting remains an interesting direction for future work. A potential avenue for future research involves characterizing a neighborhood around the optimum in which the Hessians of the loss functions can be uniformly bounded, i.e., their eigenvalues lie within $[m, L]$ for some $0 < m \le L$. This would allow us to draw on classical results from optimization theory for the class $\mathcal{S}(m, L)$ of strongly convex functions with Lipschitz-continuous gradients. In such settings, it may be possible to rigorously extend convergence guarantees and comparative analyses to the nonlinear regime. More generally, for any convergent optimization algorithm, the nonlinear dynamics necessarily approach the linearized ones asymptotically near the optimum. Thus, for any convergent algorithm, the trajectory of the nonlinear dynamics must eventually enter a neighborhood of the optimum where the linear approximation becomes accurate. Therefore, our results can also be interpreted as describing either the local or asymptotic behavior of the optimization dynamics.

Finally, although our discrete-time analysis highlights robustness advantages of natural gradient methods, it does not fully capture the stochasticity inherent in stochastic gradient descent (SGD). Developing a more precise theoretical model that explicitly incorporates the stochastic dynamics of SGD remains an important avenue for future work.

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

# A Gradient Flows for $\mathcal{L}_p^*$

For a given target distribution $p \in S_n$ and an initial distribution $q_0 \in S_n$, consider the gradient flow dynamics described by equation 46 and equation 47, and the natural gradient flow dynamics described by equation 48 given by

$$\dot{\eta}(t) = -\nabla \mathcal{L}_p^*(\eta(t)), \qquad\qquad \eta(0) = \eta_{q_0}, \tag{46}$$

$$\dot{\theta}(t) = -\nabla \mathcal{L}_p^*(\theta(t)), \qquad\qquad \theta(0) = \theta_{q_0}, \tag{47}$$

$$\dot{\theta}_{ng}(t) = -\text{grad } \mathcal{L}_p^*(\theta_{ng}(t)) = -\theta_{ng}(t) + \theta_p, \quad \theta_{ng}(0) = \theta_{q_0}. \tag{48}$$

The following is an analogue of Theorem 3 applied to the above dynamics.

**Theorem 9** (Convergence analysis)**.** *Let $p \in S_n$ and $q_0 \in S_n$ be such that $D(q_0||p)$ is sufficiently small. Suppose $\eta$, $\theta$ and $\theta_{ng}$ be the solutions to dynamics described by equation 46, equation 47 and equation 48, respectively. Then*

*(i) there exist positive constants $m_\theta^* \leq L_\theta^* < 1$, $c_\theta^*$ and $\bar{c}_\theta^*$ such that*

$$c_\theta^* e^{-2t} \leq c_\theta^* e^{-2L_\theta^* t} \leq \mathcal{L}_p^*(\theta(t)) \leq \bar{c}_\theta^* e^{-2m_\theta^* t} \qquad \forall t \geq 0, \tag{49}$$

*i.e., $\mathcal{L}_p^*(\theta(t))$ converges exponentially with rate lower than 2.*

*(ii) there exist positive constants $1 < m_\eta^* \leq L_\eta^*$ and $c_\eta^*$ such that*

$$c_\eta^* e^{-2L_\eta^* t} \leq \mathcal{L}_p^*(\eta(t)) \leq c_\eta^* e^{-2m_\eta^* t} \leq c_\eta^* e^{-2t} \qquad \forall t \geq 0 \tag{50}$$

*i.e., $\mathcal{L}_q(\eta(t))$ converges exponentially with rate higher than 2.*

*(iii) there exist positive constants $c_1^*$, $c_2^*$ and $T$ such that*

$$c_1^* e^{-2t} \leq \mathcal{L}_p^*(\theta_{ng}(t)) \leq c_2^* e^{-2t} \qquad \forall t \geq 0, \tag{51}$$

*i.e., $\mathcal{L}_p^*(\theta_{ng}(t))$ converges exponentially with rate 2.*

*Proof.* Consider the gradient flow dynamics described by equation 46. Analogous to the proof of Theorem 3 (see equation 66 and the discussion thereafter), it can be shown using the Pinsker inequality (Mohri et al., 2018, Proposition E.7) that $\eta_p$ is the unique minimizer of $\mathcal{L}_p^*$ and $S_\eta^* := \{\eta \in \phi_m(S_n)|\mathcal{L}_p^*(\eta) \leq \mathcal{L}_p^*(\eta(0))\}$ is compact. Furthermore, compactness of $S_\eta^*$ along with bound 15 implies that

$$I \prec m_\eta^* \cdot I \preceq \nabla^2 \mathcal{L}_p^*(\eta) \preceq L_\eta^* \cdot I \qquad \forall \eta \in S_\eta^*$$

where $m_\eta^* = \min_{\eta \in S_\eta^*} \lambda_{\min}\left(\nabla^2 \mathcal{L}_p^*(\eta)\right) > 1$ and $L_\eta^* = \max_{\eta \in S_\eta^*} \lambda_{\max}\left(\nabla^2 \mathcal{L}_p^*(\eta)\right)$. Applying Proposition 1, we get the desired inequality given in equation 50.

Now consider dynamics described by equation 47. Continuity of $\nabla^2 \mathcal{L}_p^*$ along with equation 18 from Lemma 2 implies that there exists an $\varepsilon > 0$ such that

$$0 \prec m_\theta^* \cdot I \preceq \nabla^2 \mathcal{L}_p^*(\theta) \preceq L_\theta^* \cdot I \prec I \qquad \forall \theta \in \mathcal{B}_\varepsilon(\theta_p) \tag{52}$$

for some positive constants $m_\theta^* \leq L_\theta^* < 1$. Using this local strong convexity condition, it can be shown that if $\mathcal{L}_p^*(\theta(0))$ is sufficiently small, $S_\theta^* := \{\theta \in \phi_e(S_n) | \mathcal{L}_p^*(\theta) \leq \mathcal{L}_p^*(\theta(0))\}$ is contained in $\mathcal{B}_\varepsilon(\theta_p)$. Applying Proposition 1, we get the desired inequality given in equation 49.

Finally consider the gradient flow dynamics described by equation 48 which can be solved exactly to obtain

$$\theta_{ng}(t) = \theta_q + e^{-t} (\theta_0 - \theta_q). \tag{53}$$

Since $\lim_{t \to \infty} \theta_{ng}(t) = \theta_q$, we can use equation 52 along with (Nesterov, 2018, Theorem 2.1.5 and Theorem 2.1.8) to show that if $\mathcal{L}_p^*(\theta(0))$ is sufficiently small, then

$$\frac{m_\theta^*}{2} ||\theta_{ng}(t) - \eta_q||^2 \leq \mathcal{L}_p^*(\theta_{ng}(t)) \leq \frac{L_\theta^*}{2} ||\theta_{ng}(t) - \theta_q||^2 \tag{54}$$

for $t \geq 0$. Plugging in the exact solution from equation 53, we get the desired inequality given in equation 51. $\qquad \square$

## B  Linearized Discrete-time Natural Gradient Dynamics in $\theta$ Coordinates

In this appendix, we show that the discrete-time natural gradient dynamics in the $\theta$ coordinates given by

$$\theta_{ng}(k+1) = \theta_{ng}(k) - \alpha_{ng} \cdot \text{grad } \mathcal{L}_q(\theta_{ng}(k)), \qquad \theta_{ng}(0) = \theta_{p_0},$$

when linearized about the equilibrium $\theta_q$, lead to update equations that are identical to the ones in the $\eta$ coordinates. From the defining property given in equation 9 of the natural gradient, and equation 6, we get that

$$\text{grad } \mathcal{L}_q(\theta) = [\nabla^2 \psi(\theta)]^{-1} \nabla \mathcal{L}_q(\theta) = [\nabla^2 \psi(\theta)]^{-1} (\nabla \psi(\theta) - \nabla \psi(\theta_q)) \approx (\theta - \theta_q),$$

where $\approx$ denotes a first-order approximation obtained by linearizing around $\theta_q$. The linearized dynamics in the $\theta$ coordinates are thus described by

$$\theta_{ng}(k+1) = \theta_{ng}(k) - \alpha_{ng} \cdot (\theta_{ng}(k) - \theta_q), \qquad \theta_{ng}(0) = \theta_{p_0},$$

which has the same form as in in equation 32.

## C  Proofs

### C.1  Proof of Proposition 1

*Proof.* [Proposition 1] Let $x$ be a solution of equation 13 and define $E(t) := f(x(t)) - f(x_*)$. Note that

$$\dot{E}(t) = \langle \nabla f(x(t)), \dot{x}(t) \rangle = -\langle \nabla f(x(t)), \nabla f(x(t)) \rangle = -||\nabla f(x(t))||^2 \leq 0. \tag{55}$$

Therefore, for all $t \geq t_0$, $f(x(t)) - f(x_*) = E(t) \leq E(t_0) = f(x(t_0)) - f(x_*)$ which implies statement $(i)$. Furthermore, we get from (Nesterov, 2018, Section 2.1) that $m \cdot I \preceq \nabla^2 f(x) \preceq L \cdot I$ for all $x \in S$ implies

$$2m(f(x) - f(x_*)) \leq ||\nabla f(x)||^2 \leq 2L(f(x) - f(x_*)) \quad \forall x \in S. \tag{56}$$

Since $x(t) \in S$ for all $t \geq t_0$, inequalities given in equation 56 and equation 55 imply that for all $t \geq t_0$,

$$-2L \cdot E(t) \leq \dot{E}(t) \leq -2m \cdot E(t).$$

Integrating from $t_0$ to $t$, we get that

$$E(t) = E(t_0) + \int_{t_0}^t \dot{E}(s)ds \le E(t_0) + \int_{t_0}^t (-2m) \cdot E(s)ds,$$

$$-E(t) = -E(t_0) + \int_{t_0}^t -\dot{E}(s)ds \le -E(t_0) + \int_{t_0}^t (-2L) \cdot (-E(s))ds$$

Finally applying the Bellman-Gronwall Lemma (Sontag, 2013, Lemma C.3.1) to the above two inequalities gives us

$$E(t) \le E(t_0)e^{-2m(t-t_0)}$$

$$-E(t) \le -E(t_0)e^{-2L(t-t_0)}$$

which dirctly gives us the desired inequality 14. $\qquad\square$

## C.2 Proof of Lemma 2

*Proof.* [Lemma 2] The Hessians of $\mathcal{L}_q$ can be evaluated using equation 5 and equation 6 as

$$\nabla^2 \mathcal{L}_q(\eta) = \nabla^2 \varphi(\eta) - D^3 \varphi(\eta)[\eta_q - \eta],$$
$$\nabla^2 \mathcal{L}_q(\theta) = \nabla^2 \psi(\theta), \tag{57}$$

where $D^3\varphi(\eta) : \mathbb{R}^n \to \mathbb{R}^{n \times n}$ is the third order derivative of $\varphi$ whose action on a vector $v \in \mathbb{R}^n$ is given by $\left(D^3\varphi(\eta)[v]\right)_{ij} = \sum_{k=1}^n \frac{\partial^3 \varphi(\eta)}{\partial \eta_i \partial \eta_j \partial \eta_k} v_k$. In particular, note that the inverse relationship given in equation 1 give us

$$\nabla^2 \mathcal{L}_q(\eta_q) = \nabla^2 \varphi(\eta_q) = [\nabla^2 \psi(\theta_q)]^{-1} = [\nabla^2 \mathcal{L}_q(\theta_q)]^{-1}. \tag{58}$$

Similarly, we can compute the Hessians of $\mathcal{L}_p^*$ using equation 7 and equation 8 as

$$\nabla^2 \mathcal{L}_p^*(\eta) = \nabla^2 \varphi(\eta), \tag{59}$$
$$\nabla^2 \mathcal{L}_p^*(\theta) = \nabla^2 \psi(\theta) - D^3 \psi(\theta)[\theta_p - \theta]$$

and use equation 1 to obtain the inverse relationship

$$\nabla^2 \mathcal{L}_p^*(\eta_p) = \nabla^2 \varphi(\eta_p) = [\nabla^2 \psi(\theta_p)]^{-1} = [\nabla^2 \mathcal{L}_p^*(\theta_p)]^{-1}. \tag{60}$$

Recall that

$$\varphi(\eta) = \left(\sum_{i=1}^n \eta_i \log \eta_i\right) + \left(1 - \sum_{j=1}^n \eta_j\right) \log\left(1 - \sum_{k=1}^n \eta_k\right)$$

and it's Hessian $\nabla^2 \varphi$ can be explicitly computed to be

$$\nabla^2\varphi(\eta) = \begin{bmatrix} \frac{1}{\eta_1} & & \\ & \ddots & \\ & & \frac{1}{\eta_n} \end{bmatrix} + \left(\frac{1}{1 - \sum_{i=1}^n \eta_i}\right) \begin{bmatrix} 1 & \cdots & 1 \\ \vdots & \ddots & \vdots \\ 1 & \cdots & 1 \end{bmatrix}. \tag{61}$$

Since the second matrix on the right hand side is positive semi-definite, we get

$$I \prec \frac{1}{\max_i \eta_i} I \preceq \nabla^2 \varphi(\eta). \tag{62}$$

This together with equation 59 gives us

$$I \prec \nabla^2 \mathcal{L}_p^*(\eta) \qquad \forall \eta \in \phi_m(S_n) \tag{63}$$

and together with equation 57 and the inverse relationship given in equation 1 gives us

$$0 \prec \nabla^2 \mathcal{L}_q(\theta) \prec I \qquad \forall \theta \in \phi_e(S_n) \tag{64}$$

proving equation 15. Furthermore, it can be shown by direct computation that

$$\nabla^2 \mathcal{L}_q(\eta) = \begin{bmatrix} \frac{[\eta_q]_1}{\eta_1^2} & & \\ & \ddots & \\ & & \frac{[\eta_q]_n}{\eta_n^2} \end{bmatrix} + \left( \frac{1 - \sum_{i=1}^n [\eta_q]_i}{\left(1 - \sum_{i=1}^n \eta_i\right)^2} \right) \begin{bmatrix} 1 & \cdots & 1 \\ \vdots & \ddots & \vdots \\ 1 & \cdots & 1 \end{bmatrix} \succ 0 \quad \forall \eta \in \phi_m(S_n). \tag{65}$$

proving equation 16.

Finally, evaluating the global bounds from equation 15 at the optimum points and using the inverse relationships given in equation 58 and equation 60 yields the desired local inequalities described by equation 17 and equation 18. □

## C.3 Proof of Theorem 3

*Proof.* [Theorem 3] First consider the gradient flow dynamics described by equation 11 and note that $\mathcal{L}_q(\theta_q) = 0$. Using the Pinsker's inequality (Mohri et al., 2018, Proposition E.7) along with the fact that $\phi_m \circ \phi_e^{-1}$ is bijective, we have that for any $\theta_p \neq \theta_q$, $\mathcal{L}_q(\theta_p) > 0$. Thus $\theta_q$ is the unique minimizer of $\mathcal{L}_q$ giving us condition $a)$ of Proposition 1. It can be shown that $\mathcal{L}_q$ has bounded sublevel sets in the $\theta$ coordinates. To see this, observe that $\|\theta\| \to \infty$ implies that for some $i$, $|\theta_i| \to \infty$, i.e., either $\theta_i \to -\infty$ or $\theta_i \to \infty$. This implies that

$$p_i = \frac{e^{\theta_i}}{1 + \sum_{j=1}^n e^{\theta_j}} \to 0 \qquad \text{or} \qquad p_{n+1} = \frac{1}{1 + \sum_{j=1}^n e^{\theta_j}} \to 0.$$

Since the $\mathcal{L}_q$ blows up to infinity on the boundary of the simplex, we get that $\mathcal{L}_q$ is coercive, i.e., $\mathcal{L}_q(\theta) = D(q\|p) \to \infty$ as $\|\theta\| \to \infty$. Observe that if some sublevel set of $\mathcal{L}_q$ is unbounded, there exists a sequence of points $\theta_{(1)}, \theta_{(2)}, \cdots$ such that $\|\theta_{(i)}\| \to \infty$ but $\mathcal{L}_q(\theta_{(i)})$ stays bounded. This is a contradiction. Therefore, we have that $\mathcal{L}_q$ has bounded sublevel sets in the $\theta$ coordinates. Finally, continuity of $\mathcal{L}_q$ implies that the sublevel sets are closed which implies that $S_\theta$ is compact. Using compactness of $S_\theta = \{\theta \in \phi_e(S_n) | \mathcal{L}_q(\theta) \leq \mathcal{L}_q(\theta(0))\}$ and the global bound given in equation 15 from Lemma 2, we get that

$$0 \prec m_\theta \cdot I \preceq \nabla^2 \mathcal{L}_q(\theta) \preceq L_\theta \cdot I \prec I \qquad \forall \theta \in S_\theta$$

where $m_\theta = \min_{\theta \in S_\theta} \lambda_{\min} \left( \nabla^2 \mathcal{L}_q(\theta) \right) > 0$ and $L_\theta = \max_{\theta \in S_\theta} \lambda_{\max} \left( \nabla^2 \mathcal{L}_q(\theta) \right) < 1$. Applying Proposition 1 gives us the desired conclusion in the form of equation 20.

Analogous to the previous case, now consider the gradient flow dynamics described by equation 10. We can use Pinsker's inequality (Mohri et al., 2018, Proposition E.7) to obtain

$$\mathcal{L}_q(\eta) \geq \frac{1}{2} \left( \left( \sum_{i=1}^n |\eta_i - [\eta_q]_i| \right) + \left| \left( 1 - \sum_{j=1}^n \eta_j \right) - \left( 1 - \sum_{k=1}^n [\eta_q]_k \right) \right| \right)^2$$

$$\geq \frac{1}{2} \left( \sum_{i=1}^n |\eta_i - [\eta_q]_i| \right)^2 = \frac{1}{2} \|\eta_q - \eta\|_1^2 \geq \frac{1}{2} \|\eta_q - \eta\|^2. \tag{66}$$

This shows that $\eta_q$ is the unique minimizer of $\mathcal{L}_q$ and together with the continuity of $\mathcal{L}$ implies that

$$S_\eta := \{\eta \in \phi_m(S_n) | \mathcal{L}_q(\eta) \leq \mathcal{L}_q(\eta(0))\}$$

is compact. Using the global bound given in equation 16, we get that

$$0 \prec \bar{m}_\eta \cdot I \preceq \nabla^2 \mathcal{L}_q(\eta) \preceq \bar{L}_\eta \cdot I \qquad \forall \eta \in S_\eta \tag{67}$$

where $\bar{m}_\eta = \min_{\eta \in S_\eta} \lambda_{\min} \left( \nabla^2 \mathcal{L}_q(\eta) \right) > 0$ and $\bar{L}_\eta = \max_{\eta \in S_\eta} \lambda_{\max} \left( \nabla^2 \mathcal{L}_q(\eta) \right)$. Applying Proposition 1, we see that there exists a positive constant $c$ such that

$$ ce^{-2\bar{L}_\eta t} \leq \mathcal{L}_q(\eta(t)) \leq ce^{-2\bar{m}_\eta t} \qquad \forall t \geq 0. \tag{68} $$

This implies that $\mathcal{L}_q(\eta(t))$ converges to 0. Thus, for any $\varepsilon > 0$, there exists a $T > 0$ such that $\mathcal{L}_q(\eta(t)) \leq 2\varepsilon$ for all $t \geq T$, which, using the lower bound obtained in equation 66 implies that $\|\eta_q - \eta(t)\| \leq \varepsilon$ for all $t \geq T$.

This means that for any $\varepsilon > 0$, there exists a $T > 0$ such that the set $S_T := \{\eta \in \phi_m(S_n) | \mathcal{L}_q(\eta) \leq \mathcal{L}_q(\eta(T))\} \subset \mathcal{B}_\varepsilon(\eta_q)$. Furthermore, continuity of $\nabla^2 \mathcal{L}_q$ along with equation 17 from Lemma 2 implies that there exists an $\varepsilon > 0$ such that

$$ I \prec m_\eta \cdot I \preceq \nabla^2 \mathcal{L}_q(\eta) \preceq L_\eta \cdot I \qquad \forall \eta \in \mathcal{B}_\varepsilon(\eta_q). $$

for some positive constants $1 < m_\eta \leq L_\eta$. Putting everything together, we get that there exists a $T \geq 0$ such that $I \prec m_\eta \cdot I \preceq \nabla^2 \mathcal{L}_q(\eta) \preceq L_\eta \cdot I$ holds for all $\eta \in S_T$. Thus, applying Proposition 1 with $t_0 = T$, we get that there exists a positive constant $\bar{c}$ such that

$$ \left( \bar{c}e^{2L_\eta T} \right) e^{-2L_\eta t} = \bar{c}e^{-2L_\eta(t-T)} \leq \mathcal{L}_q(\eta(t)) \leq \bar{c}e^{-2m_\eta(t-T)} = \left( \bar{c}e^{2m_\eta T} \right) \bar{c}e^{-2m_\eta t} $$

holds for all $t \geq T$ which is the desired form in equation 19 with $c_\eta = \bar{c}e^{2L_\eta T}$ and $\bar{c}_\eta = \bar{c}e^{2m_\eta T}$. Finally, note that if $\mathcal{L}_q(\eta(0))$ is sufficiently small, then $S_0 \subset \mathcal{B}_\varepsilon(\eta_q)$ implying that the above bound holds with $T = 0$. This completes the proof for statement (i).

Finally consider the gradient flow dynamics described by equation 12 which can be solved exactly to obtain

$$ \eta_{ng}(t) = \eta_q + e^{-t} \left( \eta_{p_0} - \eta_q \right). \tag{69} $$

Using (Nesterov, 2018, Theorem 2.1.5 and Theorem 2.1.8) along with equation 67, we get that

$$ \frac{\bar{m}_\eta}{2} ||\eta_{ng}(t) - \eta_q||^2 \leq \mathcal{L}_q(\eta_{ng}(t)) \leq \frac{\bar{L}_\eta}{2} ||\eta_{ng}(t) - \eta_q||^2. \tag{70} $$

Plugging in the exact solution from equation 69, we get the desired inequality given in equation 21. $\qquad \square$

## C.4 Proof of Theorem 4

*Proof.* [Theorem 4] Since $\nabla^2 \psi(\theta_q)$ is symmetric positive definite, we can consider its symmetric matrix square root $D_q$, such that $\nabla^2 \psi(\theta_q) = D_q \cdot D_q$. Setting $A = \sqrt{c} \cdot D_q$ in equation 22 and equation 23 and using the inverse relationship $\nabla^2 \varphi(\eta_q) = [\nabla^2 \psi(\theta_q)]^{-1}$ we obtain equation 24. Using the continuity of Hessians, we get that for any $\varepsilon > 0$, there exists a $\delta > 0$ such that

$$ (c - \varepsilon) \cdot I \preceq \nabla^2 \mathcal{L}_q(\bar{\eta}) \preceq (c + \varepsilon) \cdot I \qquad \forall \bar{\eta} \in \mathcal{B}_\delta(\bar{\eta}_q), $$

$$ \left( \frac{1}{c} - \varepsilon \right) \cdot I \preceq \nabla^2 \mathcal{L}_q(\bar{\theta}) \preceq \left( \frac{1}{c} + \varepsilon \right) \cdot I \qquad \forall \bar{\theta} \in \mathcal{B}_\delta(\bar{\theta}_q). $$

Analogous to the proof of Theorem 3, we can apply Proposition 1 to obtain the desired inequalities given in equation 25 and equation 26. $\qquad \square$

## C.5 Proof of Lemma 6

*Proof.* [Lemma 6] Recall that

$$ \nabla^2 \mathcal{L}_q(\eta_q) = \nabla^2 \varphi(\eta_q) = [\nabla^2 \psi(\theta_q)]^{-1} = [\nabla^2 \mathcal{L}_q(\theta_q)]^{-1}. $$

This establishes the equality $\text{cond}(\nabla^2 \mathcal{L}_q(\eta_q)) = \text{cond}(\nabla^2 \mathcal{L}_q(\theta_q))$. Recall that

$$ \nabla^2 \varphi(\eta) = \begin{bmatrix} \frac{1}{\eta_1} & & \\ & \ddots & \\ & & \frac{1}{\eta_n} \end{bmatrix} + \left( \frac{1}{1 - \sum_{i=1}^n \eta_i} \right) \begin{bmatrix} 1 & \cdots & 1 \\ \vdots & \ddots & \vdots \\ 1 & \cdots & 1 \end{bmatrix}. \tag{71} $$

Applying Weyl's inequalities Bhatia (2007) to the above rank one perturbation matrix, we get

$$\lambda_{\max}\left(\nabla^2\varphi(\eta)\right) \geq \frac{1}{\eta_{\min}} \qquad \text{and} \qquad \lambda_{\min}\left(\nabla^2\varphi(\eta)\right) \leq \frac{1}{\eta_{\min,2}}.$$

This directly implies the desired inequality given in equation 33. $\qquad\square$

### C.6 Proof of Theorem 7

*Proof.* [Theorem 7] Consider perturbed dynamics of the form

$$x(k+1) = x(k) - \alpha(I + \Delta(k))Q(x(k) - x^*), \tag{72}$$

where $Q$ is a symmetric positive definite matrix. This encompasses the dynamics described by equation 34, equation 35 and equation 36 by setting $Q$ equal to $\nabla^2\varphi(\eta_q)$, $\nabla^2\psi(\theta_q)$ and $I$, respectively and setting $x^*$ equal to $\eta_q$, $\theta_q$ and $\eta_q$, respectively.

Let $\kappa = \frac{\lambda_{\max}(Q)}{\lambda_{\min}(Q)}$. We will now prove that the dynamics described by equation 72 are stable, i.e., $\lim_{k\to\infty}\|x(k) - x^*\| = 0$, if for some $\varepsilon > 0$, $\|\Delta(k)\|_2 < \frac{1}{\kappa} - \varepsilon$ for all $k \geq 0$. This would directly imply statement $(i)$ by plugging in $Q = I$.

By defining the error variable $e(k) := x(k) - x^*$, we get that

$$e(k+1) = (I - \alpha Q)e(k) - \alpha\Delta(k)Qe(k). \tag{73}$$

Note that $\alpha$ is chosen optimally assuming no noise ($\Delta(k) \equiv 0$), i.e., $\alpha = \frac{2}{\lambda_{\max}(Q)+\lambda_{\min}(Q)}$ (see Theorem 5). With this choice of $\alpha$, we get that $\|I - \alpha Q\|_2 = \frac{\kappa-1}{\kappa+1}$ and $\|\alpha Q\|_2 = \frac{2\kappa}{\kappa+1}$, where the $\|\cdot\|_2$ is the induced $2-$norm which coincides in our case to the largest eigenvalue magnitude owing to symmetry. This can be seen most directly by an eigenvalue decomposition of the involved matrices. Therefore, using the triangle inequality and the submultiplicative rule of induced matrix norms, we get that,

$$\|e(k+1)\| \leq \left(\frac{\kappa-1}{\kappa+1} + \|\Delta(k)\|_2\frac{2\kappa}{\kappa+1}\right)\|e(k)\| = \frac{(1+2\|\Delta(k)\|_2)\kappa - 1}{\kappa+1}\|e(k)\|.$$

Note that if $\|\Delta(k)\|_2 < \frac{1}{\kappa} - \varepsilon$ for all $k \geq 0$, then $\frac{(1+2\|\Delta(k)\|_2)\kappa-1}{\kappa+1} < \overbrace{1 - \varepsilon\frac{\kappa}{\kappa+1}}^{\rho} < 1$ which gives us the chain of inequalities

$$\|e(k+1)\| \leq \rho\|e(k)\| \leq \rho^2\|e(k-1)\| \leq \cdots \leq \rho^{k+1}e(0).$$

Since $\rho < 1$, we get that $\lim_{k\to\infty}\|e(k)\| = 0$. Since $Q = I$ implies $\kappa = 1$, this implies statement $(i)$.

Since the above argument only derives a sufficient condition for stability, we still need to prove statement $(ii)$ separately. We now construct a time-invariant perturbation $\Delta$ such that $\|\Delta\|_2 = \frac{1}{\kappa}$ and the dynamics described by equation 72 are unstable, i.e., $x(k)$ does not converge to $x^*$. To this end, let $Q = U\Lambda U^T$ be the eigenvalue decomposition of the symmetric matrix $Q$ where $\Lambda$ is the diagonal matrix containing eigenvalues $\lambda_1 \leq \lambda_2 \leq \cdots \leq \lambda_n$ in ascending order along the diagonal. Construct $\Delta$ as

$$\Delta = U\begin{bmatrix} 0 & \cdots & 0 & 0 \\ \vdots & \ddots & \vdots & \vdots \\ 0 & \cdots & 0 & 0 \\ 0 & \cdots & 0 & \frac{1}{\kappa} \end{bmatrix}U^T.$$

Plugging this in equation 73 and using $\alpha = \frac{2}{\lambda_1+\lambda_n}$, we get that

$$e(k+1) = Me(k), \tag{74}$$

where $M = (I - \frac{2}{\lambda_1+\lambda_n}(I + \Delta)Q)$ contains an eigenvalue at $-1$. Since stability of dynamics described by equation 74 requires the spectral radius of $M$ to be less than 1, and since $M$ contains an eigenvalue at $-1$, the dynamics are unstable. This completes the proof for statement $(ii)$ by applying the constructed $\Delta$ to the choices $Q = \nabla^2\varphi(\eta_q)$ and $Q = \nabla^2\psi(\theta_q)$, respectively. $\qquad\square$

### C.7 Proof of Theorem 8

*Proof.* [Theorem 8] Following the same strategy as in the proof of Theorem 7, consider perturbed dynamics of the form

$$x(k+1) = x(k) - \alpha Q(x(k) - x^*) + \delta(k), \tag{75}$$

where $Q$ is a symmetric positive definite matrix. This encompasses the dynamics described by equation 37, equation 38 and equation 39 by setting $Q$ equal to $\nabla^2 \varphi(\eta_q)$, $\nabla^2 \psi(\theta_q)$ and $I$, respectively and setting $x^*$ equal to $\eta_q$, $\theta_q$ and $\eta_q$, respectively. By defining the error variable $e(k) := x(k) - x^*$, we get that

$$e(k+1) = (I - \alpha Q)e(k) + \delta(k). \tag{76}$$

Note that $\alpha$ is chosen optimally assuming no noise ($\delta(k) \equiv 0$), i.e., $\alpha = \frac{2}{\lambda_{\max}(Q) + \lambda_{\min}(Q)}$ (see Theorem 5). With this choice of $\alpha$, we get that $\|I - \alpha Q\|_2 = \frac{\kappa - 1}{\kappa + 1}$, where the $\|\cdot\|_2$ is the induced $2-$norm which coincides in our case to the largest magnitude eigenvalue owing to symmetry. Define $P(k) := \mathbb{E}[e(k)e(k)^T]$ where the expectation is taken over the different realizations of the noise process $\delta(k)$. Using the fact that $\delta(k)$ and $e(k)$ are independent random variables along with $\mathbb{E}[\delta(k)] \equiv 0$ and $\mathbb{E}[\delta(k)\delta(k)^T] \equiv I$, we get that

$$P(k+1) = (I - \alpha Q)P(k)(I - \alpha Q) + I. \tag{77}$$

Since $(I - \alpha Q)$ has all eigenvalues in $(-1, 1)$, it can be shown that $\lim_{k\to\infty} P(k) = P$ where $P$ solves the

$$P = (I - \alpha Q)P(I - \alpha Q) + I. \tag{78}$$

To solve this equation, let $Q = U\Lambda U^T$ be the eigenvalue decomposition of the symmetric matrix $Q$ where $\Lambda$ is the diagonal matrix containing eigenvalues $\lambda_1 \le \lambda_2 \le \cdots \le \lambda_n$ in ascending order along the diagonal. Note that equation 78 can be solved to obtain

$$P = U \begin{bmatrix} \frac{1}{1-\mu_1^2} & & 0 \\ & \ddots & \\ 0 & & \frac{1}{1-\mu_n^2} \end{bmatrix} U^T,$$

where $\mu_i$ are the eigenvalues of $(I - \alpha Q)$. Therefore, the largest eigenvalue of $P$ is $\frac{1}{1-\left(\frac{\kappa-1}{\kappa+1}\right)^2} = \frac{(\kappa+1)^2}{2\kappa}$. This proves statements $(i)$ and $(ii)$ by plugging in $Q$ equal to $\nabla^2 \varphi(\eta_q)$ and $\nabla^2 \psi(\theta_q)$, respectively. Finally, plugging $Q = I$ and $\alpha = \frac{2}{\lambda_{\min}(Q) + \lambda_{\max}(Q)} = 1$ in equation 77 directly gives statement $(iii)$. $\square$

