# OpenReview forum: "Convergence Properties of Natural Gradient Descent for Minimizing KL Divergence"
_TMLR — Accepted by TMLR_

### Review · Reviewer_Ccrj · 2025-05-25

**Summary Of Contributions:**

This paper revisits gradient flows of the (both exclusive and inclusive) KL divergence over the probabilistic simplex. To be specific, the authors studied the convergence properties of two Euclidean gradient flows as the probabilistic simplex admits two different coordinates (mixture and exponential family) and the natural gradient flow (Riemannian gradient flow) which is coordinate-independent. They showed that, for the objective being the inclusive KL, the Euclidean gradient flow w.r.t. the mixture coordinate exhibits faster convergence than the natural gradient flow, which, in turn, exhibits faster convergence than the Euclidean gradient flow w.r.t. the exponential family coordinate. They further showed that this "sandwich" relation also holds after an affine transformation of the coordinates. Analogous results were derived for the exclusive KL.

This main result shows that the common argument in the literature which is "Natural gradient descent is faster Euclidean gradient descent due to having straighter trajectories" is quite hand-waving and is not always the case. Indeed, in this study, nature gradient flow has a straight trajectory towards the optimal solution, yet converges more slowly than the Euclidean gradient flow w.r.t. the mixture coordinate.

Another contribution of the paper is the demonstration that (discrete) natural gradient descent is more robust than both Euclidean gradient descents under the presence of multiplicative and additive noise. This finding partially explains the empirically observed superior performance of natural gradient descent compared to Euclidean gradient descent in imperfect environments, such as those involving noise and discretization.

**Audience:**

Yes

**Claims And Evidence:**

Yes

**Requested Changes:**

- Proof of Theorem 3: equation (14) only implies that $L_q $ is strictly convex instead of strongly convex. Therefore, it seems to me that it is not enough to conclude the set $S_{\theta}$ is compact (mass can escape to infinity). Please clarify this point and make necessary changes with appropriate citations for the results used. Similarly, please provide more details for the argument $S_T \subset B_{\epsilon}(\eta_q)$, particularly the transition from the smallness of $L_q(\eta(T))$ to the smallness of $S_T$.

- Lemma 2 provides a global bound for the Hessian of ${L}_q$ in the $\theta$ coordinate and a local bound in the $\eta$ coordinate. In the perspective of a reader, I implicitly assume that there is no global bound in the $\eta$ coordinate in this case. However, a global bound in the $\eta$ coordinate exists  in the proof of Theorem 3 (eq. (55)). I suggest incorporating this result into Lemma 2 to make it more complete.

- Proof of Theorem 7: Here $\Delta(k)$ is time-varying so we can not define the universal $\rho$ at the end of page 20.

- Figure 4 and the sentence "...and shows why the natural gradient flow falls in between the two extremes". I do not think the reference plot of  $f(s)=(1/2) s^2$ can represent the local section of the KL divergence for the Riemannian geodesics. I suggest the authors to draw a third subfigure for the local section of the KL along geodesics.

- Page 5: the language could have been made more precise: $\langle \cdot \rangle_g$ is defined as a dot-product in the tangent spaces of $S_n$, not on $S_n$ itself. You can write "a Riemannian metric $g$ on $S_n$" instead. By the way, the tangent space of $S_n$ is not explicitly defined anywhere, please identify it more precisely.

- In Theorem 3 and other places, please say that $c_{\eta}, \bar{c}_{\eta}$, etc. are positive.

- The notations $\eta_{qi}$, etc. in Lemma 6 and other places in the Appendix are very confusing. I suggest using $[\eta_{q}]_i$, etc. instead.

### Typos:

- Equation (20), $\nabla \phi(\theta_q)$ should be  $\nabla \phi(\eta_q)$.

- Page 18, line 5 in Section D.3: $S$ should be $S_{\theta}$.

**Strengths And Weaknesses:**

## Strengths:

- The paper provides an interesting insight that the good performance of the natural gradient descent cannot easily be attributed to its straight trajectories and Euclidean gradient flows can be faster than natural gradient flow if the coordinate is well-chosen for the problem at hand. This is expected to have an impact and open up a new research direction: given an objective function, design coordinates so that Euclidean gradient flow is better than the natural gradient flow.

- The writing is of high quality, and the mathematical exposition is generally precise. Illustrative figures are provided along the way to back the claims, which I found quite pleasant to follow.

## Weaknesses:

- The faster convergence of the Euclidean gradient flow only exhibits after some positive time T that we do not have any quantitative knowledge about. This makes the conclusion "the Euclidean gradient flow w.r.t. the mixture coordinate is faster than the natural gradient flow" a bit qualitative, i.e., we do not really know when it is actually faster (can be at the very end of the flow).

- The paper lacks a discussion of how the studied gradient flows relate to machine learning contexts. For example, in Section 4, applying these flows in practice requires full knowledge of the target distribution (as reflected in $\eta_q$ and $\theta_q$), which is typically unavailable in real-world machine learning scenarios.

- The analysis in Section 4 relies on linearizing the gradient flows around the optimal solution (i.e., the target distribution). Again, it assumes access to the very distribution we aim to approximate. Furthermore, it remains unclear how one can rigorously derive concrete results for the original (nonlinear) dynamics based on this linearized analysis.

- Some mathematical arguments need a careful scrutiny, as detailed in my Requested Changes below.

---

> ### Author Response · Authors · 2025-06-04
>
> Thank you very much for your thoughtful and constructive review. We truly appreciate the time and care you took in engaging with our work. We are following a two-stage revision process. In this first stage, we have focused on implementing all requested changes and typos you suggested along with addressing the first item in the weakness section, viz., that the faster convergence of the Euclidean flow occurs only after some unknown time $T$, making the result qualitative. These are summarized in the comments below in a point-by-point basis. As for items 2 and 3 from the weaknesses section concerning the relationship of our work to machine learning practice and the limitations of linearized analysis: we fully agree these are important issues. We are currently conducting additional empirical experiments aimed at bridging the gap between the theoretical results and realistic machine learning scenarios. These new results will be included in our second-stage update. We would be very happy to receive any further feedback you may have on the current set of changes and welcome any clarifying questions or suggestions on points already revised.

---

> ### Author Response · Authors · 2025-06-04
> **Response to the first three requested changes**
>
> - Regarding Proof of Theorem 3: Thank you for your careful reading. You are absolutely correct that strict convexity alone does not, in general, imply that the level sets of a function are bounded (i.e., compactness is not guaranteed in general). Our argument relied on the fact that if a strictly convex function achieves its minimum at a point $x^* \in \mathbb{R}^n$, then its level sets are indeed bounded. While this holds for our situation, we agree that an explicit argument is needed to ensure clarity. We have revised the proof of Theorem 3 to clarify this point and to provide a more rigorous justification.Specifically,
>     - We now explicitly argue that for our setting, the KL divergence $L_q(\theta)$ grows unbounded as $\lVert\theta\rVert \rightarrow \infty$. This ensures the boundedness of sublevel sets in the $\theta$ coordinates without appealing solely to strict convexity. This clarification is included in the first paragraph of the proof of Theorem 3.
>      - In addition, although not originally requested, we also included an analogous argument in the $\eta$ coordinates for completeness. By leveraging Pinsker's inequality, we show directly that small values of $L_q(\eta)$ imply that $\eta$ lies within a small neighborhood of the target $\eta_q$. This both establishes boundedness of the level sets in the $\eta$ coordinates and justifies the inclusion $S_T \subset B_{\varepsilon}(\eta_q)$ rigorously. These additions are highlighted in red around equation (57).
> - Regarding Lemma 2: We appreciate this helpful suggestion. Following your recommendation, we have revised Lemma 2 to explicitly include the global positive definiteness of the Hessian $\nabla^2 L_q(\eta)$. This now reflects the result used later in the proof of Theorem 3 (Equation (55)) and makes the lemma more complete and self-contained. In addition, we have clarified that a similar global bound does not exist in the $\theta$ coordinates. To support this, we included a counterexample immediately following Lemma 2, which demonstrates that the level sets of $L_q(\theta)$ are non-convex, implying the nonexistence of a global lower bound on the Hessian in the $\theta$ parameterization. This discussion was previously in the appendix but has now been moved into the main text in response to your suggestion, which we agree enhances the paper’s clarity and accessibility.
> - Regarding Proof of Theorem 7: Thank you for pointing out this inconsistency. You are absolutely right. Our original argument incorrectly used a universal constant $\rho$ while $\Delta(k)$ is in fact time-varying. This oversight stems from an earlier version of the work in which we considered time-invariant perturbations, and we did not fully update the argument when generalizing to the time-varying case. We apologize for the confusion. We have now revised Theorem 7 and its proof to address this issue. Specifically, we have replaced the assumption $\lVert \Delta(k)\rVert <1$ for all $k\geq 0$ with the requirement that there exists an $\varepsilon>0$, arbitrarily small, such that $\lVert \Delta(k) \rVert < 1-\varepsilon$. This revised assumption ensures uniform boundedness away from $1$, which allows us to establish exponential convergence with a fixed contraction rate. The resulting proof is cleaner and more robust, while still capturing the desired convergence behavior. We are grateful for your careful reading and helpful comment, which significantly improved the rigor of this result.

---

> ### Author Response · Authors · 2025-06-04
> **Response to the last four requested changes and Typos**
>
> - Regarding Figure 4: Thank you for your thoughtful comment. We appreciate your suggestion to include a third sub-figure showing the local behavior of the KL divergence along Riemannian geodesics. Our intention with Figure 4 was to provide geometric intuition into convergence behaviors by visualizing local cross-sections of the loss function near the optimum along various directions. Specifically, we plot $s\mapsto L_q(\eta_q+s\cdot v_i)$ for multiple directions $v_i$ on the unit circle in the $\eta$ coordinates, and similarly $s\mapsto L_q(\theta_q+s\cdot w_i)$ in the $\theta-$coordinates. While these curves incidentally correspond to the $m-$ and $e-$geodesics, our primary intention was not to make a statement about geodesics per se, but to contrast the local curvature and behavior of the loss landscape in different parameterizations. Although it would be interesting in principle to include a cross-section along a Riemannian geodesic induced by the Levi-Civita connection, we feel this would extend beyond the illustrative purpose of this figure. If we have misunderstood your comment, we would happy to understand this better. We also understand that this figure was not discussed to a sufficient detail in the original manuscript. In response, we have added a clarifying explanation on page 9. In particular, we now explicitly state in the text that the natural gradient dynamics, governed by Equation (12), can be equivalently described as $\dot{\eta}=-\nabla f_q (\eta(t))$ where $f_q(\eta(t))=\frac{1}{2} \lVert \eta_{q} - \eta(t)\rVert^2$. This shows that the natural gradient flow corresponds to gradient descent on a quadratic function with identity Hessian, justifying our use of $f(s)=\frac{1}{2}s^2$ as a proxy for the loss landscape. With this clarification, we believe the figure effectively conveys the intended contrast between the loss landscapes underlying the $\theta$, $\eta$, and natural gradient dynamics. Your feedback helped us sharpen the explanation, and we thank you again for the constructive suggestion.
> - Regarding the notation for Riemannian metric on page 5: Thank you for this helpful suggestion. We have updated the language in the text above Equation (9) to clarify that $\langle \cdot,\cdot \rangle_p$ is the inner product defined on the tangent spaces and corresponds to the Riemannian metric $g$ on $S_n$. In addition, we now explicitly identify the tangent space $T_pS_n$ while providing a citation to standard reference for further details on the geometry of the simplex. We believe this improves the clarity and precision of the exposition and we appreciate your feedback.
> - Regarding the constants $c_{\eta}$ and $c_{\bar{\eta}}$: We have adopted your suggestion. Thank you.
> - Regarding the notation for $\eta_{q_i}$: We have adopted your suggestion. Thank you
> - Regarding the two typos: We have fixed these errors. Thank you.

---

> ### Author Response · Authors · 2025-06-04
> **Response to the first weakness, viz., the bound holds only for $t\geq T$**
>
> Thank you for this insightful observation. We have revised the manuscript to clarify and more transparently discuss the nature of the convergence rate comparison in the following ways:
> - We have added a sentence to the statement of Theorem 3 specifying that if $L_q(\eta(0))$ is sufficiently small, the result holds with $T=0$. This formalizes the intuition that the bound becomes effective immediately when the initial point lies sufficiently close to the target.
> - Following Theorem 3, we have added a paragraph elaborating on the asymptotic nature of the convergence result. In particular, we note that while the bound is guaranteed only after some (generally unknown) time $T>0$, this is a common feature in the analyses of asymptotic convergence rates. We also provide a sufficient condition under which the bound holds with $T=0$, namely that $ \nabla L_q (\eta) \succ I $ for all $\eta$ belonging to the sublevel set $S_{\eta}:=\{\eta\in \phi_m(S_n)|L_{q}(\eta)\leq L_{q}(\eta(0))\}$.
> - We highlight that empirical results show that the convergence rate ordering predicted by theory tends to emerge early in the flow (i.e., $T$ is small in practise), supporting the practical relevance of the asymptotic comparison.
> - Finally, we have also revised the discussion of Theorem 9, which addresses the dual objective $L_p^*$, to streamline the presentation and directly accommodate the $T=0$ scenario.

---

> ### Author Response · Authors · 2025-06-13
> **Response to the second and partly third weakness: assumption of full knowledge of the target distribution**
>
> We thank the reviewer for raising this important concern. In response, we have significantly revised the manuscript further in the second revision to address the practical limitations of assuming full knowledge of the target distribution. Specifically, we have added a new section (Section 5) titled "Numerical Experiments with Empirical KL Divergence", which explicitly bridges the gap between our theoretical framework and practical machine learning settings. In this section, we:
>
> 1. **Define and motivate the empirical KL divergence** $\hat{\mathcal{L}}_{\mathcal{D}}$, which is computed from a finite dataset $\mathcal{D}$ of i.i.d. samples from the target distribution $q$ and we comment on how the results developed with the knowledge of $q$ can be connected to this data-based setting.
>
> 2. **Incorporate comprehensive simulations in this empirical setting** (Figures 8–13 and Tables 1–2) that investigate the behavior of gradient descent and stochastic gradient descent when optimizing empirical objectives. These experiments demonstrate that
>     - The sandwiching property continues to hold for small and equal learning rates,
>     - NGD consistently outperforms GD in terms of convergence speed when learning rates are individually optimized.
>     - These results hold across both low- and high-dimensional settings.
>     - These results hold across both full-batch (gradient descent) and stochastic gradient descent settings  .
> 5. **Summarize all findings in a comparison table** (Table 3), highlighting the consistency between theoretical results and empirical observations.
>
> We believe this new section substantively addresses the reviewer's concern by demonstrating that our theoretical insights are not only of conceptual value, but also directly applicable to real-world scenarios where only samples from the target distribution are available. We hope this strengthens the overall contribution of our work and clarifies its practical relevance.

---

> ### Author Response · Authors · 2025-06-13
> **Response to the third weakness: reliance on linearization around the optimum in Section 4**
>
> We thank the reviewer for this concern. We agree that extending the analysis beyond the linearized regime is an important and meaningful direction for future research. In the current version, our discrete-time theoretical results are indeed based on a local linearization of the gradient flows around the optimal distribution. This linearization enables us to derive clean, interpretable convergence guarantees and facilitates direct comparisons between different update dynamics. We also note that for any convergent optimization method, the trajectory will eventually enter a neighborhood of the optimum where the linear approximation becomes valid. Thus, our theoretical results can be interpreted as characterizing the local or asymptotic behavior of the optimization dynamics.
>
> More importantly, although the theoretical results in Section 4 rely on linearized dynamics, the newly added Section 5 goes beyond this setting empirically. The simulated dynamics in that section are fully nonlinear, and we consider both full-batch and stochastic gradient descent settings using empirical loss functions derived from sampled data. These results demonstrate that the key insights from our theoretical analysis continue to hold in more realistic scenarios, at least empirically.
>
> While a full nonlinear theoretical analysis is beyond the scope of this work, one promising direction involves identifying a neighborhood around the optimum where the Hessian eigenvalues of the loss function are uniformly bounded between $m$ and $L$. In such regions, classical analysis techniques for functions in the class $\mathcal{S}(m, L)$ (m-strongly convex with L-Lipschitz gradients) could be used to extend our convergence guarantees to the nonlinear setting within a specified region around the optimum.
>
> We have added a discussion summarizing these points in Section 6 of the revised manuscript. We hope this clarifies how our current contributions connect to both practical implementations and directions for future theoretical extensions.

---

> > ### Comment · Reviewer_Ccrj · 2025-06-20
> >
> > Thank you for the detailed and thoughtful answers to my questions and concerns. With the additional discussions and experiments, I believe the manuscript is now in strong shape -- mathematically sound and well connected to the machine learning community.
> >
> > Best regards,
> > Reviewer Ccrj

---

### Review · Reviewer_rhRm · 2025-05-26

**Summary Of Contributions:**

From the viewpoint of information geometry, Categorical distributions can be parameterized using either the exponential family parameterization or the mixture family parameterization. The paper studies the convergence properties of gradient descent (GD) under these two different parameterizations and natural gradient descent (NGD). It was shown that, in the continuous time setting NGD not necessarily outperforms GD; instead, its convergence rate may lie between two parameterizations of GD. However, under discrete time setting, NGD performs better and yields more robust updates.

**Audience:**

Yes

**Broader Impact Concerns:**

Not applicable as the paper is theoretical

**Claims And Evidence:**

Yes

**Requested Changes:**

Can the authors comment on the applicability of the analysis to more complex cases, e.g. for distributions that are not in the exponential family or when training a non-linear neural network?

Minor:

Page 9, third to last time of the last paragraph of Section 3: "discrete-tine" should be "discrete-time"

**Strengths And Weaknesses:**

Strengths: 1. The research problem is fundamental in nature. While one can argue that NGD is natural, its convergence properties are less clear. 2. The paper provided interesting and rigorous observations in those regards.

Weaknesses: 1. The paper focuses on learning a Categorical distribution. While it is an interesting scenario, it is to some extent limited, and it is unclear whether the results apply for more general scenarios.

---

> ### Author Response · Authors · 2025-06-04
>
> Thank you very much for taking the time to review our submission and for your encouraging and constructive feedback. We have addressed both of your requested changes. In particular, we have corrected the typographical error on page 9. Additionally, we have added a brief discussion in the second and third paragraphs of Section 5 to address the scope of our analysis, as well as potential directions for generalization. In particular, we now comment on how the current theoretical results derived for categorical distribution may extend in several meaningful directions:
>  - Beyond exact natural gradient: Theorem 7 already accommodates certain structured perturbations of the natural gradient update, suggesting that convergence guarantees may still hold when the Fisher information matrix is approximated (e.g., through methods like K-FAC), provided these approximations meet norm bounds ensuring stability. We view this as a step toward analyzing practical optimization algorithms that approximate natural gradient.
> - Beyond the probability simplex and KL divergence: While our analysis is centered on the simplex with dual coordinate systems induced by the KL divergence, we highlight that the tools employed can plausibly extend to more general dually flat manifolds which are statistical manifolds equipped with general Bregman divergences beyond KL. We point out that under appropriate curvature conditions, it may be possible to establish analogues of our main results, such as the convergence rate ordering in Theorem 3.
> - Beyond exponential families: Finally, we point out that mixtures of exponential families with hidden variables such as Boltzmann machines (which are no longer exponential families themselves) is another candidate research direction and exploring convergence behavior in such models could be promising for future work.

---

### Review · Reviewer_DB1o · 2025-05-31

**Summary Of Contributions:**

The paper revisits natural-gradient descent (NGD) for minimising KL divergence on the simplex and delivers four take-aways:
- Continuous-time theory: NGD’s convergence rate is sandwiched between two Euclidean gradient flows—slower than mixture-coordinate GD but faster than exponential-coordinate GD—so it is not universally optimal.
- Coordination insight: An affine re-parameterisation can always make a Euclidean gradient flow match NGD’s rate, underscoring that NGD’s edge lies in its coordinate invariance, not raw speed.
- Discrete-time advantage: Once discretised, NGD enjoys perfect conditioning (Hessian condition number = 1), giving faster linear convergence and greater robustness to gradient noise than the Euclidean updates.
- Empirical confirmation: Simple 2-component mixture experiments align closely with the analytic rates and visualise why NGD’s straight geodesic paths are sometimes slower in continuous time yet superior in discrete settings.

**Audience:**

Yes

**Broader Impact Concerns:**

I believe this work does not raise any ethical concerns, as it is a theoretical study focused on the fundamental issue of convergence of (N)GD in the context of machine learning.

**Claims And Evidence:**

Yes

**Requested Changes:**

The following list is what I believe it is important to clarify and correct. If I have misunderstood anything, I would greatly appreciate your guidance in identifying the inaccuracies.

- Extend the empirical study beyond $n=2$.
  - Please include experiments on at least one higher-dimensional simplex (e.g., $n=5 \ \textrm{or} \ 10$) and report the observed linear-rate constants and stability margins, and discuss whether the “η-GD ≻ NGD ≻ θ-GD” ordering and the $κ = 1$ conditioning advantage of NGD still hold.

- More detailed limitation of this work and future direction
  - The manuscript would benefit from a more detailed discussion of the applicability and limitations of the presented theoretical findings, as well as important future directions. For example, it would be valuable to clarify that the current analysis does not account for practical approximations such as K-FAC, and to speculate on how such approximations might affect the convergence behavior or robustness properties. Such a discussion would help a broader TMLR readership better understand the scope and implications of the work.
  - Briefly comment on other Bregman or f-divergences: would the sandwich ordering or $κ = 1$ property plausibly generalise? Even a speculative remark with citations would help position the work.

- Correction of Typos.
  - Please carefully check for typographical errors throughout the paper (e.g., Section 3).

**Strengths And Weaknesses:**

First and foremost, I would like to express my sincere appreciation for the authors’ efforts in developing this paper.
I would also like to note that the following review is written with the understanding that TMLR places a strong emphasis on accuracy, persuasiveness, and clear evidence, as well as capturing the interest of readers, rather than solely focusing on novelty or impact.


## Strengths
- **Rigorous, transparent theory:** The paper derives exponential-rate bounds for the three flows and proves the “sandwich” ordering in Theorem 3, resting on uniform Hessian eigenvalue bounds (Lemma 2).
- **Fresh coordinate-invariance perspective:** The affine-reparameterisation result (Theorem 4) shows how Euclidean GD can be made arbitrarily faster or slower while NGD stays fixed at rate 2, clarifying that NGD’s edge is invariance, not intrinsic speed.
- **Discrete-time insight and robustness:** By linking each method to a quadratic with Hessian condition number $κ$, the authors show NGD enjoys $κ = 1$ and therefore the best linear rate and the largest stability margin; Lemma 6 and Theorem 7 back this up for multiplicative and additive noise.
- **Empirical corroboration:** Trajectories and log-slope fits for $n = 2$ mixtures match the analytic predictions almost exactly (slopes ≈ 7 / 2.0 / 0.5 for η-GD/NGD/θ-GD in Figure 3.), giving the theory convincing face validity.
- **Clear presentation:** The exposition is largely self-contained, with helpful figures, consistent notation and full proofs in the appendix, making the paper accessible to both optimisation and information-geometry audiences.


## Weaknesses
- **Limited empirical scope:** All experiments use $2$-dimensional simplices; it remains unclear whether the continuous-time ranking or the $κ = 1$ advantage of NGD persists in higher-dimensional or real-world models.
- **Theoretical evaluation relying on single metric:** Results are specific to KL divergence (I understand that this should place as a future work); extending the analysis to other Bregman or $f$-divergences—or even squared loss on the simplex—would will push the better understanding of convergence of NGD.
- **Limitation of noise model analyses:** In robustness analysis, the authors assume a GD scenario with either deterministic relative noise or i.i.d. additive noise. Therefore, the theoretical results presented in this paper do not yet extend to more realistic and commonly used settings—such as those involving stochastic gradients—where both noise types may coexist and exhibit state-dependent variance. Extending the analysis to such contexts remains an important direction for future work.
- **Limitation of theoretical understanding:** The theoretical results presented in this paper do not address the convergence behavior when using practical natural gradient approximations such as K-FAC, nor do they provide detailed analysis for alternative geometry-aware optimization methods, such as mirror descent. This could be also an important direction for future work.
- **Typos:** A few minor issues, e.g., “discrete-tine” → “time” (in Section 3).

## Concerns regarding the interest of TMLR readers
- While the paper delivers a neat information-geometric analysis, its scope is narrowly focused on KL divergence over the probability simplex and on the natural-vs-Euclidean gradient debate.  Many TMLR readers—especially those working on large-scale model or applications—may not immediately perceive actionable take-aways because the toy-scale examples stop at $n=2$ components. Of course, using the setting $n = 2$ is entirely reasonable for verifying the theoretical results. However, including results for multiple values of n would make the findings more intuitive and enhance their credibility.

These points aside, the paper offers a crisp theoretical contribution with a nuanced take on when natural gradients help, and with a bit more empirical breadth and practical discussion it could be a strong addition to TMLR.

---

> ### Author Response · Authors · 2025-06-04
>
> Thank you very much for your thoughtful and constructive review. We truly appreciate the time and care you took in engaging with our work. We are following a two-stage revision process. In this first stage, we have focused on implementing the requested changes and typos. We are currently conducting additional empirical experiments aimed at bridging the gap between the theoretical results and realistic machine learning scenarios. These new results will be included in our second-stage update. We would be very happy to receive any further feedback you may have on the current set of changes and welcome any clarifying questions or suggestions on points already revised. Thank you again for your detailed and generous review.

---

> ### Author Response · Authors · 2025-06-04
> **Response to requested changes**
>
> - Regarding the empirical study beyond $n=2$: Thank you for this valuable suggestion. We have extended our empirical study to the case $n=10$ as recommended. The corresponding results are now presented in Figure 6 and Figure 7. These new experiments confirm that the qualitative behavior observed in the $n=2$ case continues to hold: the convergence rate ordering $\eta-$GD $>$ NGD $>$ $\theta-$ GD persists. We discuss these findings in the second-to-last paragraph of page 9 in the revised manuscript. We also make the interesting observation based on these empirical studies that the difference in convergence rates is more pronounced for $n = 10$ as compared to $n = 2$. Regarding the discrete-time experiments, we are currently conducting more detailed studies including the case for $n=10$. We plan to include updated results in our next revision as discussed above and we are grateful for your patience and encouragement on this front.
>
> - Regarding the detailed limitation of this work and future direction: Thank you for your thoughtful suggestions regarding the limitations and broader applicability of our theoretical findings. We have incorporated a detailed discussion addressing these points in the second and third paragraph of Section 5. This includes clarifying the scope of our analysis with respect to exact natural gradient descent, commenting on the potential impact of practical Fisher matrix approximations such as K-FAC, and speculating on how the key structural properties might extend to general Bregman divergences and dually flat manifolds. We hope this provides additional clarity and better situates our contributions within a broader optimization and information geometry context.
>
> - Regarding the Typos: These have been fixed. Thank you.

---

> ### Author Response · Authors · 2025-06-13
> **Response to concerns on empirical scope and limitations of noise model analyses**
>
> We thank the reviewer for these thoughtful comments highlighting the limitations of the original submission. We have substantially revised the manuscript to address these concerns.
>
> #### 1. **Extended empirical scope**
>
> We have gone beyond the setting of $n=2$ and significantly expanded the empirical study (see Section 5), including:
> - Simulations in higher dimensions, e.g., $n=10$ as suggested.
> - Systematic discrete-time simulations comparing empirical convergence times of natural gradient descent (NGD) and standard gradient descent (GD) in both $\eta$ and $\theta$ coordinate systems.
> - Analyses over variations in the learning rates and obtained convergence behaviors.
>
> The extended experiments confirm that the theoretical findings continue to hold under more realistic optimization settings.
>
> #### 2. **Empirical extension to stochastic gradient descent**
>
> Thank you for pointing this out. Yes, the robustness results in Section 4 focused on simplified noise models, viz., either deterministic relative perturbations or additive i.i.d. noise, and these did not cover the full complexity of stochastic gradients encountered in practice.
>
> To address this, we have added new simulations in Section 5.1 involving stochastic gradient descent (SGD), where gradients are computed over randomly drawn minibatches from the data. These experiments:
> - Use empirical KL divergence computed from minibatches as the optimization objective.
> - Include both full-batch GD and mini-batch SGD dynamics.
> - Demonstrate that the qualitative behavior predicted by our theory continues to hold empirically under stochastic updates.
>
> We agree that developing a rigorous theoretical extension to formally analyze the general stochastic settings is a valuable direction for future work. We now explicitly highlight this as a key future research direction in Sections 6 of the revised manuscript.
>
> We hope these revisions address the reviewer’s concerns and help demonstrate the broader relevance of our theoretical contributions in more realistic and practical optimization settings.

---

> > ### Comment · Reviewer_DB1o · 2025-06-15
> > **Acknowledgement**
> >
> > Dear authors,
> >
> > I thank the authors for their detailed and thoughtful responses. The revisions and additions---especially the additional experiments and discussions of limitation, future direction, and relevance with approximation methods---substantially improve the practical contribution of the paper, in my opinion.
> > I believe the paper makes a solid and rigorous theoretical contribution; I think this is good paper.
> >
> > Sincerely,
> >
> > --Reviewer DB1o

---

### Decision · Action_Editor_RURg · 2025-07-16

**Recommendation:** Accept as is

**Audience:**

Yes

**Audience Explanation:**

To quote from two different reviewers, the work concerns results that are "Fundamental in nature" and "Taken together, the work adds a nuanced perspective on when natural gradients matter and provides both theoreticians and practitioners with actionable insights."

**Claims And Evidence:**

Yes

**Claims Explanation:**

The paper mostly focuses on mathematical arguments, and after the revision, the reviewers are satisfied with the correctness of the proofs.  The writing is sound and clear.  There are also some small-dimensional numerical examples.